# Cell and circuit origins of fast network oscillations in the mammalian main olfactory bulb

**Shawn D Burton[1,2]\*[†], Nathaniel N Urban[1,2]\*[†]**

[1]Department of Neurobiology, University of Pittsburgh, Pittsburgh, United States; [2]Center for the Neural Basis of Cognition, Pittsburgh, United States

**Abstract** Neural synchrony generates fast network oscillations throughout the brain, including the main olfactory bulb (MOB), the first processing station of the olfactory system. Identifying the mechanisms synchronizing neurons in the MOB will be key to understanding how network oscillations support the coding of a high-dimensional sensory space. Here, using paired recordings and optogenetic activation of glomerular sensory inputs in MOB slices, we uncovered profound differences in principal mitral cell (MC) vs. tufted cell (TC) spike-time synchrony: TCs robustly synchronized across fast- and slow-gamma frequencies, while MC synchrony was weaker and concentrated in slow-gamma frequencies. Synchrony among both cell types was enhanced by shared glomerular input but was independent of intraglomerular lateral excitation. Cell-type differences in synchrony could also not be traced to any difference in the synchronization of synaptic inhibition. Instead, greater TC than MC synchrony paralleled the more periodic firing among resonant TCs than MCs and emerged in patterns consistent with densely synchronous network oscillations. Collectively, our results thus reveal a mechanism for parallel processing of sensory information in the MOB via differential TC vs. MC synchrony, and further contrast mechanisms driving fast network oscillations in the MOB from those driving the sparse synchronization of irregularly firing principal cells throughout cortex.

**\*For correspondence:**
shb420@lehigh.edu (SDB);
nnu220@lehigh.edu (NNU)

**Present address:** [†]Department of Biological Sciences, Lehigh University, Bethlehem, United States

**Competing interest:** The authors declare that no competing interests exist.

## Introduction

Fast network oscillations are widespread in neural activity throughout the mammalian brain, including the main olfactory bulb (MOB), where gamma-frequency (~40–100 Hz) oscillations reflecting the synchronous firing of principal cells are intimately linked with olfactory learning, memory, and behavior (*Martin and Ravel, 2014*). Identifying the mechanisms underlying gamma-frequency synchronization of principal cells in the MOB will be key to understanding how fast network oscillations contribute to the neural coding of a complex, high-dimensional sensory space (*Uchida et al., 2014*). While decades of experimental and modeling studies have identified an important contribution of lateral inhibitory circuits to the synchronization of principal mitral cells (MCs) (*Rojas-Líbano and Kay, 2008*), more complete mechanistic understanding is limited by at least three gaps in knowledge.

First, whether and how tufted cells (TCs) synchronize their firing has not been tested. Overwhelming evidence has established that TCs, a second type of excitatory MOB principal cell, differ from MCs in their intrinsic and synaptic properties, sensory responses, and axonal projections (*Shepherd et al., 2004*; *Nagayama et al., 2014*; *Burton et al., 2020*). Not only do these findings support a model in which TCs and MCs form parallel pathways encoding complementary information, but they further suggest that TCs and MCs may differentially engage in fast network oscillations. In particular, weaker lateral inhibition among TCs than MCs (*Christie et al., 2001*; *Geramita et al., 2016*) suggests that TCs may synchronize less than MCs.

Second, how gamma-frequency oscillations separately emerge across fast (~60–100 Hz) and slow (~40–60 Hz) frequency bands remains unknown. Fast- and slow-gamma-frequency oscillations in the MOB are differentially modulated by state (*Kay, 2003*; *Frederick et al., 2016*; *Zhuang et al., 2019*), suggesting both behavioral relevance and at least partially distinct underlying sources. Indeed, the temporal sequencing of fast- and slow-gamma-frequency oscillations across early and late phases of the sniff cycle (*Lepousez and Lledo, 2013*; *Manabe and Mori, 2013*; *Frederick et al., 2016*), paralleling early and late sensory-evoked TC and MC firing, has motivated the attractive but untested hypothesis that TCs and MCs synchronize across fast- and slow-gamma frequencies, respectively (*Manabe and Mori, 2013*; *Mori et al., 2013*).

Finally, how lateral inhibitory circuits support gamma-frequency oscillations in the MOB remains unclear. A large population of granule cell (GC) interneurons mediate lateral inhibition among MOB principal cells, and several studies have proposed that periodic GC-mediated inhibition opens windows of opportunity for a subset of MCs to synchronously fire across a sparse fraction of gamma-frequency cycles (i.e., a sparsely synchronous oscillation or 'sparse synchrony') (*Rall and Shepherd, 1968*; *Eeckman and Freeman, 1990*; *Neville and Haberly, 2003*; *Bathellier et al., 2006*; *Schoppa, 2006*), paralleling pyramidal-interneuron gamma (PING) theories elsewhere in the brain (*Wang, 2010*; *Buzsáki and Wang, 2012*). Alternative theories, however, instead point toward the capacity of correlated synaptic currents, independent of periodicity, to reset the phase of resonant neural oscillators, synchronizing periodic firing across several consecutive gamma-frequency cycles (i.e., 'dense synchrony') (*Desmaisons et al., 1999*; *Galán et al., 2006*; *Rubin and Cleland, 2006*; *David et al., 2015*).

Here, we used paired cell-type-specific recordings in acute MOB slices together with optogenetic stimulation of sensory inputs to investigate the cell and circuit origins of fast network oscillations in the MOB. Under conditions mimicking the odor-evoked firing patterns of TCs and MCs observed in vivo, TCs exhibited robust, widespread, and enduring spike-time synchrony across fast- and slow-gamma frequencies, while MC synchrony was weaker and largely concentrated in slow-gamma frequencies. Greater synchronization further emerged between cells with convergent rather than divergent glomerular inputs, but occurred independent of lateral excitation, which was absent among TCs. Within both MCs and TCs, spike-time synchronization correlated with firing periodicity, while surprisingly neither excitatory nor inhibitory synaptic currents exhibited detectable gamma-frequency patterning. These results, together with the observation of greater intrinsic resonance among TCs than MCs, argue that gamma-frequency oscillations in the MOB emerge in large part from the dense synchronization of periodic firing among resonant TCs – findings with critical implications for the encoding and propagation of olfactory information.

## Results

### Multiglomerular activation evokes greater gamma-frequency spike-time synchrony among TCs than MCs

To investigate the cell and circuit origins of fast network oscillations in the MOB, we recorded TC pairs and MC pairs in acute slices prepared from OMP-ChR2:EYFP mice while photostimulating olfactory sensory neuron (OSN) terminals in glomeruli at 5 Hz to mimic the physiological dynamics of sniff-paced sensory input (*Wachowiak, 2011*). As odorants frequently activate clusters of glomeruli in a concentration-dependent manner (*Mori et al., 2006*), we used full-field photostimulation to activate OSN terminals within multiple neighboring glomeruli. Such multiglomerular activation evoked ~20 Hz firing in MCs on average (*Figure 1—figure supplement 1A–D*), with MCs of a pair occasionally firing synchronously ($|\Delta t_{spike}| \leq 5$ ms) (*Figure 1A and D*), similar to previous investigation of spike-time synchrony in MC pairs using electrical OSN stimulation (*Schoppa, 2006*). Under identical conditions, multiglomerular activation evoked more rapid TC firing (*Figure 1—figure supplement 1E–H*) and a remarkable degree of TC spike-time synchrony (*Figure 1G and J*). Consistent with visual inspection of cell-attached traces, spike-time cross-correlograms among TCs exhibited prominent central peaks ($|\Delta t_{spike}| \leq 5$ ms) compared to minimal peaks among MCs (*Figure 1B, E, H and K*).

TC spike-time cross-correlograms also exhibited slow-timescale modulation (*Figure 1H and K*), reflecting the more phasic firing of TCs than MCs (*Figure 1—figure supplement 1*). Cell-type differences in cross-correlogram central peaks may thus emerge from this difference in phasic vs. tonic

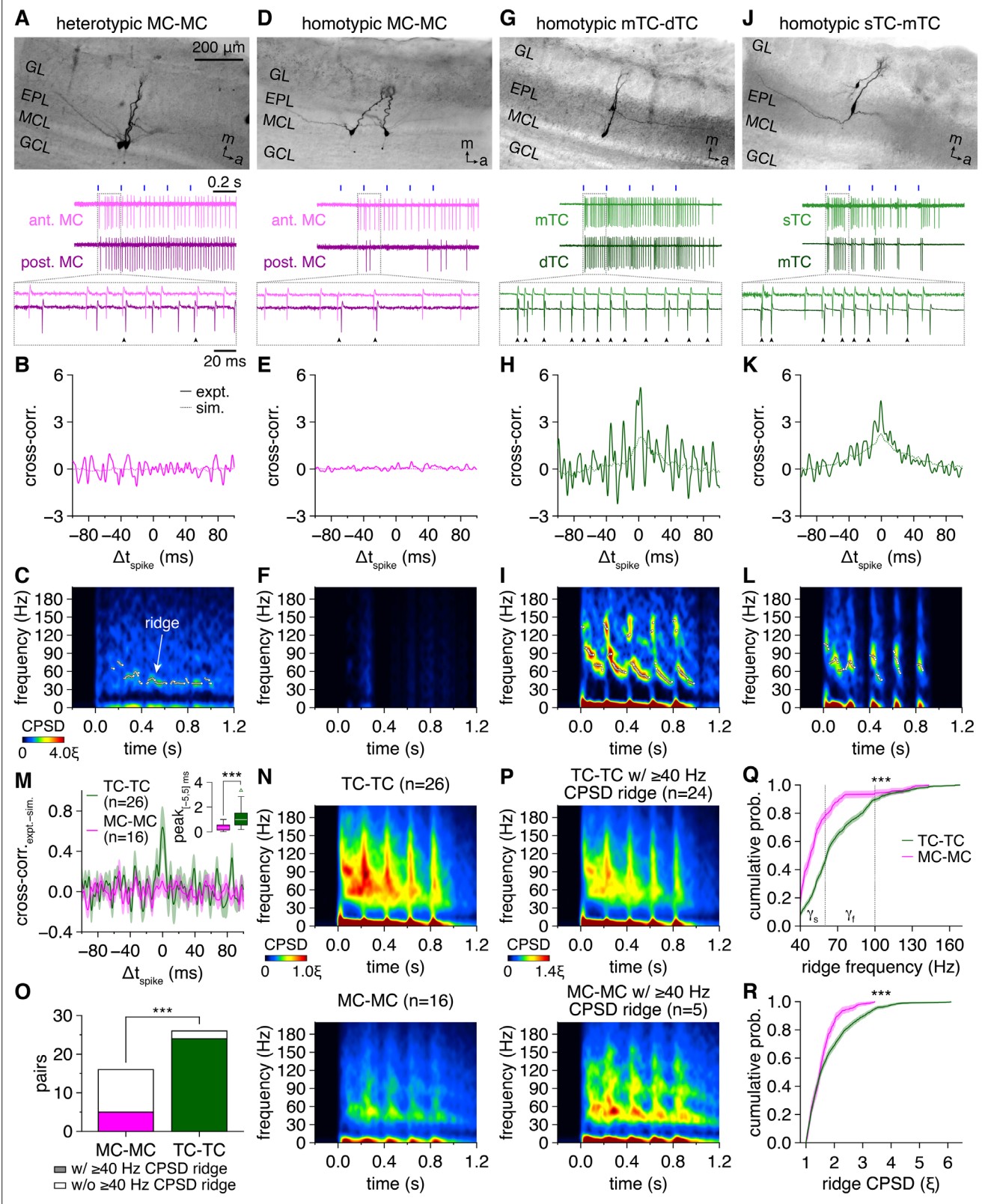

**Figure 1.** Multiglomerular activation evokes widespread synchronization of tufted cell (TC) firing across fast- and slow-gamma frequencies and limited synchronization of mitral cell (MC) firing across slow-gamma frequencies. (**A**) Example cell-attached recording of a heterotypic MC pair during photostimulation. Morphology (upper) and representative trial (lower; blue rectangles: 10 ms light pulses of the 5 Hz photostimulation protocol; arrowheads: synchronous spikes, $|\Delta t_{spike}| \leq 5$ ms). GL: glomerular layer; EPL: external plexiform layer; MCL: mitral cell layer; GCL: granule cell layer; m:

*Figure 1 continued on next page*

*Figure 1 continued*

medial; a/ant.: anterior; post.: posterior. (**B**) Trial-averaged cross-correlogram of spike times recorded throughout the photostimulation protocol in the MC pair in (**A**) ('expt.'), compared to the cross-correlogram of spike times simulated from rate-matched independent Poisson processes ('sim.'). (**C**) Trial-averaged spike-time cross-power spectral density (CPSD) spectrogram from the MC pair in (**A**) following photostimulation onset at 0.0 s. Continuous epochs (ΔHz/ms < 150) of high CPSD reflecting robust periodic synchrony are defined as 'ridges' and demarcated with white circles. Color is scaled by multiples of the ridge threshold ( $\xi$ ). (**D–L**) Same as (**A–C**) for a homotypic MC pair (**D–F**), a homotypic middle TC (mTC) and deep TC (dTC) pair (**G–I**), and a homotypic superficial TC (sTC) and mTC pair (**J–L**). Scaling in (**F, I, L**) is the same as in (**C**). (**M**) Mean spike-time cross-correlograms (with slow-timescale firing rate correlations removed via subtraction of the simulated spike-time cross-correlograms) of TC pairs and MC pairs. Inset: cross-correlogram peaks within $|\Delta t_{spike}| \leq 5$ ms were higher among TCs than MCs (Wilcoxon rank-sum test: p=3.0 × 10$^{-4}$). (**N**) Spike-time CPSD spectrograms averaged across all TC pairs (upper) and MC pairs (lower). (**O**) More TC than MC pairs exhibited spike-time CPSD ridges (chi-squared test: p=3.2 × 10$^{-5}$, $\chi^2$ = 17.3). (**P**) Spike-time CPSD spectrograms averaged across all TC pairs (upper) and MC pairs (lower) exhibiting CPSD ridges. (**Q, R**) Cumulative distributions of frequencies (**Q**) and CPSD (**R**) across all spike-time CPSD ridges. TCs exhibited faster (**Q**) (two-sample Kolmogorov–Smirnov test: p=1.7 × 10$^{-44}$) and more precise (**R**) (two-sample Kolmogorov–Smirnov test: p=3.2 × 10$^{-11}$) gamma-frequency synchrony than MCs. Shading denotes 95% confidence intervals. γ$_s$: slow-gamma frequencies, 40–60 Hz; γ$_f$: fast-gamma frequencies, 60–100 Hz.

The online version of this article includes the following figure supplement(s) for figure 1:

**Figure supplement 1.** Multiglomerular activation evokes higher firing rates and more phasic firing patterns among tufted cells (TCs) than mitral cells (MCs).

**Figure supplement 2.** Mitral cell (MC) pairs and tufted cell (TC) pairs exhibit greater gamma-frequency spike-time synchrony than spike times simulated from rate-matched independent Poisson processes.

**Figure supplement 3.** Tufted cells (TCs) exhibit greater gamma-frequency spike-time synchrony than mitral cells (MCs) on both a per-pair basis and independent of ridge-based analyses.

**Figure supplement 4.** Fast-gamma-frequency spike-time synchrony emerges early in each 5 Hz photostimulation cycle and decelerates toward slow-gamma frequencies.

firing alone, rather than a difference in network-driven spike-time synchrony. Excluding this possibility, however, simulated spike trains generated from independent Poisson processes with rates matching experimental firing rate patterns (*Figure 1—figure supplement 1*) replicated the slow-timescale correlations observed in TC spike times but failed to replicate the prominent cross-correlogram central peaks (*Figure 1B, E, H and K*). Indeed, isolating fast-timescale synchrony exceeding chance levels (by subtracting simulated from experimental spike-time cross-correlograms) revealed markedly higher central peaks among TCs than MCs (*Figure 1M*). Multiglomerular activation thus evokes greater spike-time synchrony among TCs than MCs independent of firing rate differences.

Spike-time cross-correlograms further exhibited pronounced periodicity manifest in regular side peaks across TC (*Figure 1H and K*) and some MC pairs (*Figure 1B*) but absent from simulated spike-time cross-correlograms. To directly investigate such periodic synchrony, we examined spike-time cross-power spectral density (CPSD) (i.e., the power spectrum of the cross-correlogram), which likewise revealed striking cell-type differences (*Figure 1C, F, I and L*). Specifically, TC pairs exhibited consistently higher CPSD levels overall, indicative of more precise spike-time synchrony. Moreover, TC firing synchronized across both fast- and slow-gamma frequencies, often with a sweeping deceleration across each photostimulation cycle, while MC synchrony was less dynamic and largely limited to slow-gamma frequencies. As with the cross-correlogram analysis of synchrony irrespective of periodicity, these differences in gamma-frequency spike-time synchrony were independent of differences in phasic vs. tonic firing patterns, as chance levels of periodic synchrony in simulated spike trains were concentrated in sub-gamma frequencies (*Figure 1—figure supplement 2*). Across all pairs, cell-type differences in the precision and frequency of periodic spike-time synchrony were profound (*Figure 1N*).

To quantify these differences, continuous epochs of periodic spike-time synchrony were isolated by detecting maximal ridges within CPSD spectrograms (*Figure 1C*), similar to previous investigations of fast network oscillations in the MOB and elsewhere (*Roux et al., 2007*; *Cenier et al., 2009*; *David et al., 2009*; *Fourcaud-Trocmé et al., 2011*; *David et al., 2013*; *David et al., 2015*). Analysis was restricted to 40–200 Hz to specifically investigate fast-timescale synchrony, and the threshold for ridge detection ( $\xi$ ) was set to the 95th percentile of 40–200 Hz CPSD values observed throughout the photostimulation protocol in all pairs, ensuring that ridges reflect epochs of robust periodic synchrony. With this approach, 92% of recorded TC pairs exhibited ≥1 CPSD ridge compared to only 31% of MC pairs (*Figure 1O*). The cell-type differences in CPSD spectrograms thus emerge at least partially

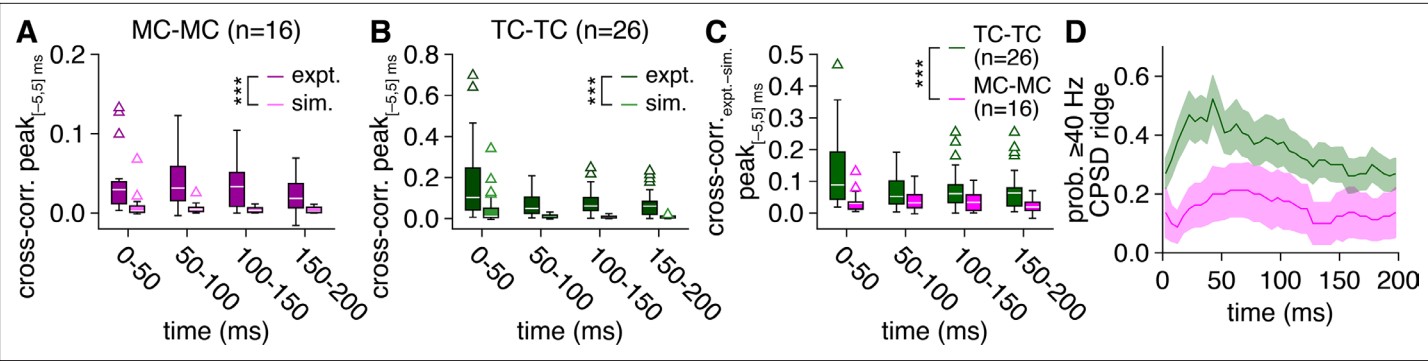

**Figure 2.** Greater synchronization of tufted cell (TC) than mitral cell (MC) firing persists throughout the average 5 Hz photostimulation cycle. (**A**) Experimental spike times recorded across MC pairs ('expt.') within consecutive 50 ms windows of the 5 Hz photostimulation cycle exhibited consistently higher cross-correlogram peaks (within $|\Delta t_{spike}| \leq 5$ ms) than spike times simulated from rate-matched independent Poisson processes ('sim.') (two-way ANOVA on ranks, expt./sim. × 50 ms window: significant main effect of expt./sim., p=1.2 × 10⁻⁹, $F_{1,120} = 43.6$; no significant main effect of 50 ms window, p=0.34, $F_{3,120} = 1.1$; no significant interaction, p=0.99, $F_{3,120} = 0.03$). (**B**) Experimental spike times recorded across TC pairs ('expt.') likewise exhibited consistently higher cross-correlogram peaks than spike times simulated from rate-matched independent Poisson processes ('sim.') (two-way ANOVA on ranks, expt./sim. × 50 ms window: significant main effect of expt./sim., p=1.2 × 10⁻¹⁹, $F_{1,200} = 102.1$; no significant main effect of 50 ms window, p=0.34, $F_{3,200} = 1.1$; no significant interaction, p=0.62, $F_{3,200} = 0.6$). (**C**) Spike-time cross-correlograms (with slow-timescale firing rate correlations removed via subtraction of simulated spike-time cross-correlograms) within consecutive 50 ms windows of the 5 Hz photostimulation cycle exhibited consistently higher peaks (within $|\Delta t_{spike}| \leq 5$ ms) among TC than MC pairs (two-way ANOVA on ranks, cell type × 50 ms window: significant main effect of cell type, p=4.6 × 10⁻⁴, $F_{1,160} = 12.8$; no significant main effect of 50 ms window, p=0.23, $F_{3,160} = 1.5$; no significant interaction, p=0.90, $F_{3,160} = 0.2$). (**D**) The probability of robust periodic spike-time synchrony reflected in spike-time cross-power spectral density (CPSD) ridges was consistently higher among TC than MC pairs throughout the average photostimulation cycle.

The online version of this article includes the following figure supplement(s) for figure 2:

**Figure supplement 1.** Full-field photostimulation evokes comparable excitatory currents in tufted cells (TCs) and mitral cells (MCs).

from more widespread periodic synchrony among TCs than MCs. Even when restricting our analysis to only those pairs exhibiting CPSD ridges, however, fundamental differences remained (*Figure 1P*), with TCs exhibiting both more precise and higher frequency periodic synchrony than MCs (*Figure 1Q and R, Figure 1—figure supplement 3A*). Examination of spike-time CPSD averaged throughout the photostimulation protocol and independent of ridge detection likewise supported these findings (*Figure 1—figure supplement 3B and C*).

Of note, while we searched for CPSD ridges across a wide frequency range, epochs of periodic spike-time synchrony were nevertheless identified almost exclusively within gamma frequencies (*Figure 1Q*), highlighting the marked tuning of MOB circuitry to gamma-frequency oscillations. Additionally, while individual pairs exhibited variable ridge dynamics, periodic synchronization at fast-gamma frequencies consistently emerged early in each photostimulation cycle, and decelerated toward slower gamma frequencies at mean rates up to 0.1–0.2 Hz/ms (*Figure 1—figure supplement 4*).

As a caveat, it is possible that the differences observed in TC vs. MC spike-time synchrony reflect the artificial conditions of our experimental preparation rather than cell-type differences in network-driven synchronization poised to shape sensory processing in vivo. Specifically, it is possible that the strong optogenetic stimulus combines with the more effective sensory input and greater excitability of TCs than MCs (*Gire et al., 2012*; *Burton and Urban, 2014*; *Jones et al., 2020*) to instantaneously synchronize TC firing (i.e., stimulus-driven synchronization). Indeed, TC firing frequently exhibited rapid synchronization following photostimulation. However, even under a barrage of predominantly asynchronous inhibitory synaptic input (see below) and following pauses in firing in one or both cells of a pair (e.g., *Figure 1G and J*), both TC and MC spike-time synchrony persisted at levels higher than expected by chance throughout the average photostimulation cycle (i.e., up to 200 ms following photostimulation) (*Figure 2A and B*). Cross-correlogram and CPSD ridge analyses further demonstrated that greater TC than MC spike-time synchrony likewise persisted throughout the entire photostimulation cycle (*Figure 2C and D*). Collectively, these results are inconsistent with stimulus-driven synchronization, which in the absence of network-driven synchronization should decay rapidly under ongoing network activity (see also discussion in *Schoppa, 2006*). Of further note, both the rates and

temporal patterning of firing recorded closely match the odor-evoked firing observed in morphologically confirmed MCs and TCs in vivo (*Nagayama et al., 2004*; *Igarashi et al., 2012*; *Phillips et al., 2012*) (and see Discussion), confirming that our optogenetic approach recapitulates key aspects of MOB sensory processing.

Consistent with the cell-type differences in spike-time synchrony reflecting real features of MOB sensory processing rather than an artificially strong activation of TC sensory input, voltage-clamp recordings obtained from a large subset of pairs following cell-attached recordings additionally revealed no cell-type difference in excitatory current amplitude and even modestly greater excitatory charge transferred to MCs than TCs throughout the photostimulation protocol (*Figure 2—figure supplement 1A–C*). Moreover, latencies from photostimulation onset to excitatory input were similar across cells of each pair among both MCs and TCs but across all pairs were fairly broadly distributed across ~10–25 ms (*Figure 2—figure supplement 1D and E*), consistent with our optogenetic approach triggering more gradual and physiological glomerular activation than single or short bursts of electrical stimuli (*Carey et al., 2009*; *Burton and Urban, 2015*). Attenuation of our optogenetic stimulus by limited light penetrance into the tissue contributed to such gradual glomerular activation, with photostimulation routinely failing to activate glomeruli deep in the slice (data not shown). Excitatory input was also completely devoid of any gamma-frequency patterning (*Figure 2—figure supplement 1F*), further arguing that the periodic spike-time synchrony observed was not directly driven by the stimulus. Of additional note, the overall comparable excitatory currents observed between cell types indicate that our optogenetic stimulus exceeded minimal effective OSN stimulation levels, whereupon glomeruli transition from all-or-none activation – with greater TC than MC input (*Gire et al., 2012*; *Burton and Urban, 2014*) – to graded activation and excitatory input (*Geramita and Urban, 2017*; *Jones et al., 2020*), consistent with our modeled scenario of a suprathreshold-concentration odorant activating a cluster of glomeruli.

Glomerular organization can significantly influence the synchronization of MC firing: irregular 0–10 Hz firing evoked by step-current injection or bath NMDA application occurs synchronously among MCs with apical dendrites converging in the same glomerulus (i.e., homotypic MCs) but asynchronously among MCs linked to different glomeruli (i.e., heterotypic MCs). This aperiodic spike-time synchrony is driven by electrical coupling of dendritic AMPAR-mediated autoreceptor potentials within the glomerulus (*Schoppa and Westbrook, 2002*; *Christie et al., 2005*), producing reliable lateral excitation between homotypic MCs (*Schoppa and Westbrook, 2002*; *Urban and Sakmann, 2002*; *Christie et al., 2005*; *Pimentel and Margrie, 2008*; *Maher et al., 2009*). Whether lateral excitation likewise promotes gamma-frequency spike-time synchrony among homotypic MCs – or even exists among homotypic TCs – is unknown, though connexin36 knockout (which abolishes both electrical coupling and glutamatergic excitation among homotypic MCs) attenuates fast network oscillations in the MOB (*Pouille et al., 2017*). Any differences in glomerular organization or lateral excitation among the MCs and TCs in our dataset may thus contribute to the observed cell-type differences in gamma-frequency spike-time synchrony.

While our dataset indeed included more homotypic TC than MC pairs, the proportion of homotypic to heterotypic pairs did not significantly differ between cell types (*Figure 3A*). Moreover, clear cell-type differences in periodic spike-time synchrony remained even when restricting our analysis to homotypic or heterotypic pairs alone (*Figure 3B*). Gamma-frequency spike-time synchrony thus fundamentally differs between MCs and TCs independent of glomerular organization.

Spike-time CPSD ridges were detected in both homotypic and heterotypic pairs at comparable rates and across largely overlapping frequencies (*Figure 3C and D*), further arguing that the cell-type differences in gamma-frequency spike-time synchrony were not driven by differences in glomerular organization. Synchrony was, however, markedly stronger across homotypic than heterotypic pairs (*Figure 3B and E*). Surprisingly, this enhancement of spike-time synchrony occurred independent of intraglomerular lateral excitation, as homotypic TCs – which displayed robust periodic spike-time synchrony (*Figure 1G and J*, *Figure 3B*) – exhibited no lateral excitation, while 100% of the homotypic MC pairs tested exhibited typically asymmetric lateral excitation matching previous accounts (*Figure 3F–H*). Lack of lateral excitation among TCs was not due to cell-type differences in electrical coupling, however, as homotypic MCs and homotypic TCs exhibited comparable electrical coupling coefficients (*Figure 3I*), as previously reported (*Ma and Lowe, 2010*). Our data thus support the hypothesis that some aspect of intraglomerular signaling – likely including electrical coupling

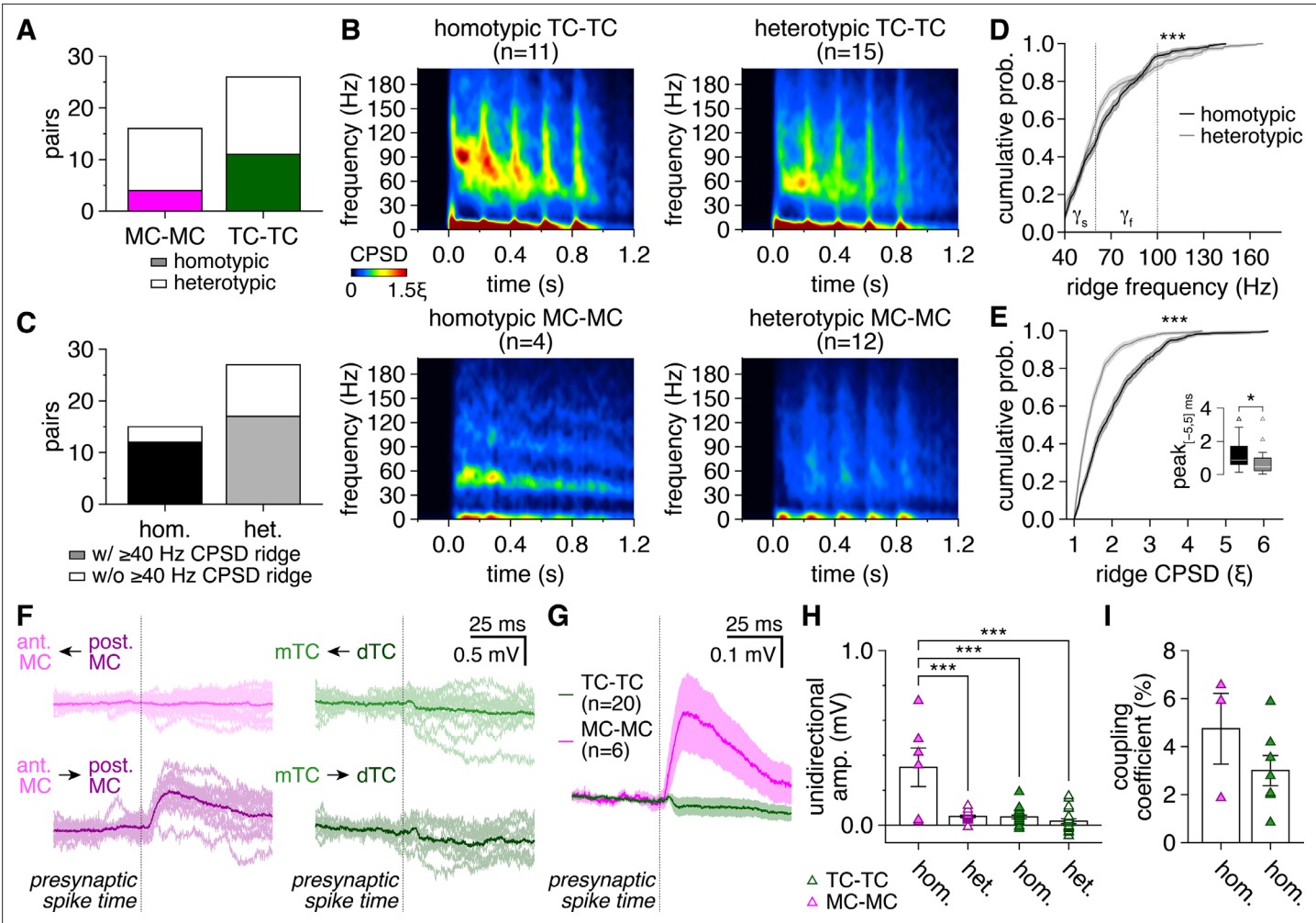

**Figure 3.** Greater synchronization of tufted cell (TC) than mitral cell (MC) firing extends across distinct patterns of glomerular organization and is not driven by intraglomerular lateral excitation. (**A**) MCs and TCs included comparable proportions of homotypic and heterotypic pairs (chi-squared test: p=0.26, $\chi^2$ = 1.3). (**B**) Spike-time cross-power spectral density (CPSD) spectrograms averaged across all homotypic (left) and heterotypic (right) TC pairs (upper) and MC pairs (lower) following photostimulation onset at 0.0 s. (**C**) Pairs with spike-time CPSD ridges were detected among both homotypic pairs ('hom.') and heterotypic pairs ('het.') at comparable rates (chi-squared test: p=0.25, $\chi^2$ = 1.3). (**D, E**) Cumulative distributions of frequencies (**D**) and CPSD (**E**) across all spike-time CPSD ridges for homotypic and heterotypic pairs. Epochs of periodic synchrony were distributed across significantly different, though largely overlapping, frequencies among homotypic and heterotypic pairs (**D**) (two-sample Kolmogorov–Smirnov test: p=3.8 × 10⁻⁷), while periodic synchrony was substantially more precise among homotypic than heterotypic pairs (**E**) (two-sample Kolmogorov–Smirnov test: p=1.6 × 10⁻⁵¹). Shading denotes 95% confidence intervals. Inset: consistent with more precise synchrony, spike-time cross-correlogram peaks within |Δt_{spike}| ≤ 5 ms (as in *Figure 1M*) were higher among homotypic than heterotypic pairs (Wilcoxon rank-sum test: p=0.033). (**F**) Unitary postsynaptic responses to single presynaptic spikes in the MC pair (left) and TC pair (right) in *Figure 1D and G*, revealing reliable asymmetric lateral excitation selectively between the homotypic MCs. Arrows: direction of transmission tested. Light traces: individual trials; dark traces: average. (**G**) Mean unitary postsynaptic response to single presynaptic spikes across homotypic MC pairs and homotypic TC pairs, revealing consistent intraglomerular lateral excitation among MCs and no visible excitation among TCs. (**H**) Lateral excitation (typically asymmetric within pairs) was exclusively detected among homotypic MCs, as revealed by stronger unitary postsynaptic response amplitudes across homotypic MC pairs than across either TC pairs or heterotypic MC pairs (two-way ANOVA, cell type × glomerular organization: significant main effect of cell type, p=1.4 × 10⁻⁷, $F_{1,60}$ = 35.6; significant main effect of glomerular organization, p=2.4 × 10⁻⁷, $F_{1,60}$ = 33.9; significant interaction, p=7.4 × 10⁻⁶, $F_{1,60}$ = 24.1; post-hoc Tukey–Kramer test: homotypic MC-MC vs. heterotypic MC-MC, p=1.7 × 10⁻⁷; homotypic MC-MC vs. homotypic TC-TC, p=6.2 × 10⁻⁸; homotypic MC-MC vs. heterotypic TC-TC, p=8.5 × 10⁻⁹; heterotypic MC-MC vs. homotypic TC-TC, p=1.0; heterotypic MC-MC vs. heterotypic TC-TC, p=0.80; homotypic TC-TC vs. heterotypic TC-TC, p=0.83). (**I**) Homotypic MC pairs and homotypic TC pairs exhibited comparable electrical coupling coefficients (two-sample t-test: p=0.22, $t_8$ = 1.3).

The online version of this article includes the following figure supplement(s) for figure 3:

**Figure supplement 1.** Profound cell-type differences in spike-time synchrony cannot be explained by differences in within-pair intersomatic distance.

**Figure supplement 2.** Tufted cell (TC) spike-time synchrony does not correlate with within-pair intersomatic differences in external plexiform layer (EPL) depth.

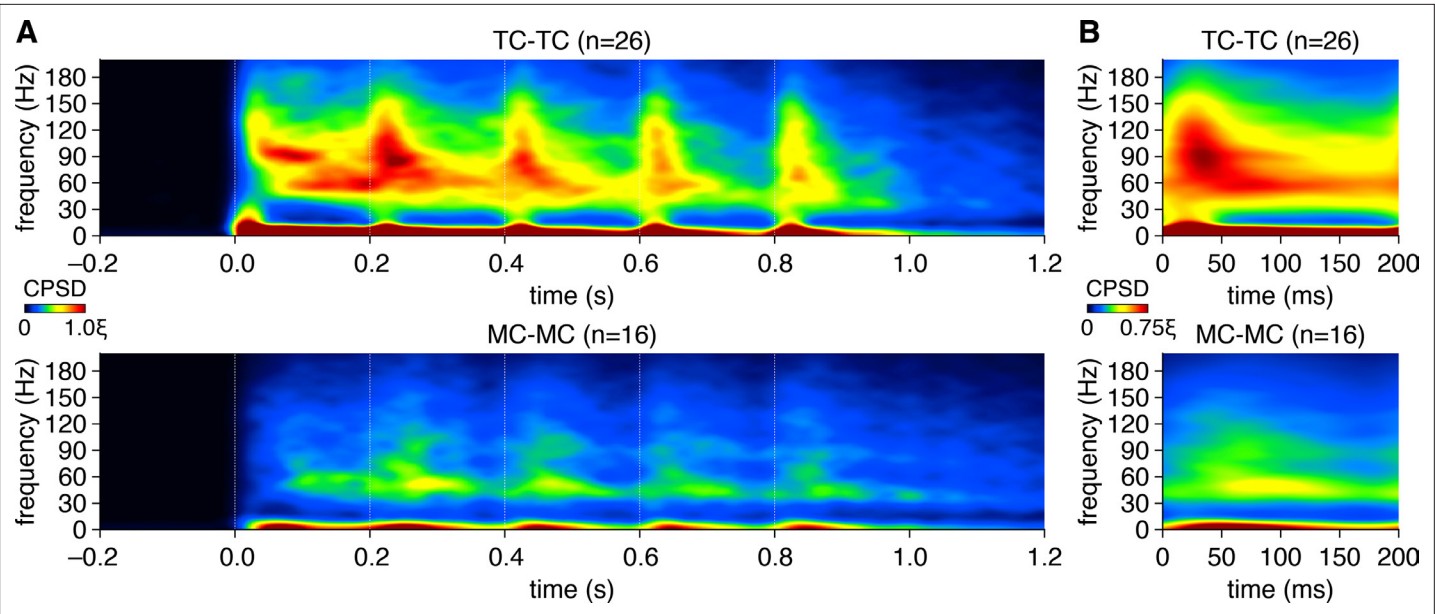

**Figure 4.** Mitral cells (MCs) and tufted cells (TCs) exhibit distinct patterns of gamma-frequency spike-time synchrony across the 5 Hz photostimulation cycle. (**A**) Spike-time cross-power spectral density (CPSD) spectrograms averaged across all TC pairs (upper) and MC pairs (lower) expanded in time across consecutive photostimulation cycles (dashed lines). Scaling is identical to *Figure 1N*. (**B**) Same as (**A**), averaged across all photostimulation cycles.

– enhances fast network oscillations in the MOB (*Pouille et al., 2017*), and further suggest that lateral excitation between homotypic MCs does not enhance (and may even hinder) periodic spike-time synchrony (see Discussion).

Differences in intersomatic distance across TC pairs and MC pairs may also contribute to cell-type differences in spike-time synchrony, given distance-dependent declines in MC and TC lateral inhibitory signaling (*Christie et al., 2001*; *Egger and Urban, 2006*) and coincidence (*Schmidt and Strowbridge, 2014*; *Arnson and Strowbridge, 2017*). TC pairs in our dataset indeed exhibited significantly shorter lateral intersomatic distances (i.e., distances parallel to the MCL) than MC pairs, despite equal total intersomatic distances (*Figure 3—figure supplement 1A and B*). Total and lateral intersomatic distance failed to correlate with cross-correlogram or CPSD measures of spike-time synchrony among TCs, however, while MCs exhibited only a modest reduction in spike-time CPSD levels with increasing lateral intersomatic distance (*Figure 3—figure supplement 1C–F*). While thus highlighting the strong lateral organization of circuitry contributing to fast network oscillations in the MOB, these results more broadly argue against a pivotal contribution of differences in intersomatic distances to the pronounced cell-type differences observed in spike-time synchrony. Similarly, while we recorded from TCs spanning the full depth of the external plexiform layer (EPL), differences in somatic depth also failed to correlate with spike-time synchrony among TCs (*Figure 3—figure supplement 2*).

In summary, our results thus demonstrate that multiglomerular activation evokes robust, widespread, and enduring synchronization of TC firing across fast- and slow-gamma frequencies and limited synchronization of MC firing primarily across slow-gamma frequencies, revealing fundamental cell-type differences that emerge across multiple analyses and cannot be explained by experimental or anatomical factors. As a caveat, the limited spike-time synchrony detected among MCs in our dataset constrains extensive characterization of the frequency of periodic MC synchronization. Our results thus do not exclude a contribution of MC spike-time synchrony to fast-gamma-frequency oscillations in the MOB. However, our results do definitively identify robust periodic spike-time synchrony among TCs as a major contributor to fast- and slow-gamma-frequency oscillations in the MOB, as reflected by the marked resemblance of TC spike-time CPSD spectrograms to LFP spectrograms recorded in the MOB of freely behaving rodents (compare *Figure 4* to Figure 2 of *Manabe and Mori, 2013*).

In addition to pronounced gamma-frequency synchrony, TC firing also exhibited substantially greater synchronization across theta frequencies than MC firing (*Figure 5*). Our results thus additionally suggest that sensory information specifically encoded in the cross-frequency coupling of MOB

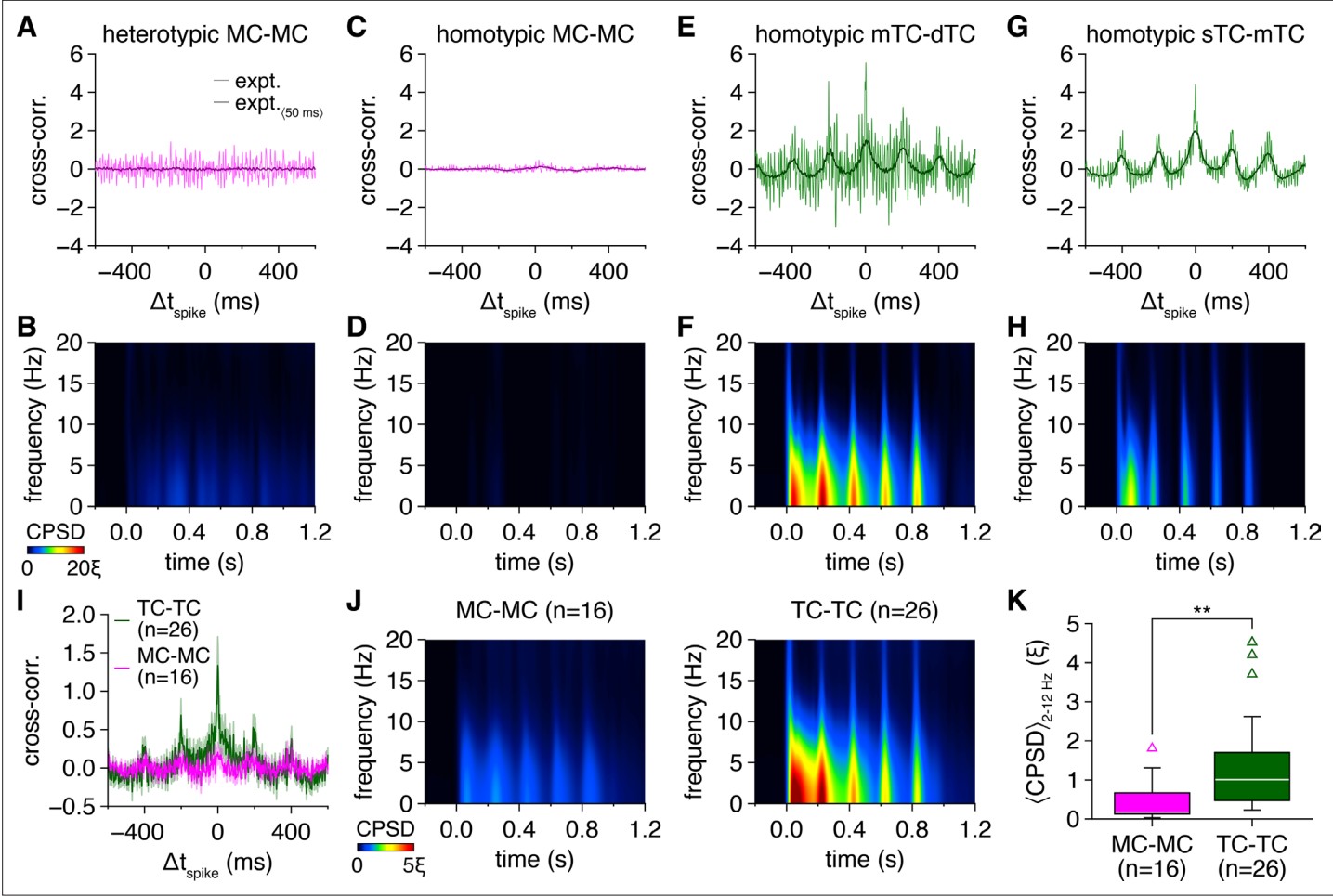

**Figure 5.** Tufted cells (TCs) exhibit greater spike-time synchrony across theta frequencies than mitral cells (MCs). (**A**) Expanded-timescale cross-correlogram of spike times recorded in the MC pair in *Figure 1A*, with 50-ms-long sliding window average applied to highlight temporal patterning at frequencies < 20 Hz. (**B**) Spike-time cross-power spectral density (CPSD) spectrogram of spike times recorded in the MC pair in *Figure 1A*, rescaled to theta frequencies. (**C–H**) Same as (**A, B**) for the MC pair and TC pairs in *Figure 1D, G and J*. (**I**) Mean expanded-timescale cross-correlograms of TC pairs and MC pairs. (**J**) Spike-time CPSD spectrograms averaged across all MC pairs (left) and TC pairs (right), rescaled to theta frequencies. (**K**) TC pairs exhibited greater spike-time CPSD averaged throughout the photostimulation protocol and across 2–12 Hz than MC pairs (Wilcoxon rank-sum test: p=1.1 × 10⁻³).

The online version of this article includes the following figure supplement(s) for figure 5:

**Figure supplement 1.** Multiglomerular activation at different theta frequencies evokes qualitatively similar patterns of periodic spike-time synchrony.

gamma- and theta-frequency oscillations (*Buonviso et al., 2006*; *Schaefer and Margrie, 2007*; *Mori et al., 2013*; *Zhong et al., 2017*; *Tort et al., 2018*; *Heck et al., 2019*; *Losacco et al., 2020*) is conveyed to downstream olfactory cortical areas by synchronous TC firing. Moreover, while we did not systematically examine photostimulation across other theta frequencies associated with rodent olfaction, in a subset of TC pairs we confirmed that 2 Hz and 8 Hz photostimulation evoked qualitatively similar patterns of periodic spike-time synchrony as 5 Hz photostimulation (*Figure 5—figure supplement 1*), suggesting that the observed cell-type differences in spike-time synchrony generalize across different olfactory sampling strategies.

## Greater synchronization of TC than MC firing is not driven by more synchronous synaptic inhibition among TCs than MCs

Identifying the mechanisms supporting differential synchronization of TC and MC firing will be critical to understanding how fast network oscillations in the MOB contribute to sensory processing. A leading theory of gamma-frequency synchrony in the MOB, supported by extensive biophysical

modeling (*Rall and Shepherd, 1968*; *Bathellier et al., 2006*; *Fourcaud-Trocmé et al., 2011*; *Pouille et al., 2017*), asserts that synchronous synaptic inhibition mediated by reciprocal MC-GC interactions temporally gates MC activity, opening windows of opportunity for a subset of MCs to synchronously fire across periodic gamma-frequency cycles (*Eeckman and Freeman, 1990*; *Neville and Haberly, 2003*; *Shepherd et al., 2004*; *Schoppa, 2006*; *Rojas-Líbano and Kay, 2008*). By this theory, cell-type differences in the synchronization of synaptic inhibition should parallel cell-type differences in periodic spike-time synchrony. We therefore hypothesized that multiglomerular activation evokes (1) greater synchronization of synaptic inhibition among TCs than MCs, and (2) dynamic synchronization of synaptic inhibition among TCs across fast- and slow-gamma frequencies and stable synchronization of synaptic inhibition among MCs across slow-gamma frequencies.

To test these hypotheses, we recorded outward inhibitory postsynaptic currents (IPSCs) in a new set of TC pairs and MC pairs using the same optogenetic approach. Multiglomerular activation evoked a prolonged barrage of inhibitory input to both MCs and TCs (*Figure 6A, D, G and J*, *Figure 6—figure supplement 1*), with increases in IPSC rate and amplitude and a decrease in IPSC decay constant (*Figure 6—figure supplement 2*). Though IPSC kinetics can influence fast network oscillations in the MOB (*Lagier et al., 2007*; *Lepousez and Lledo, 2013*), no differences in IPSC rise-time or decay constant were observed between cell types (*Figure 6—figure supplement 2F and G*). Evoked IPSC rates were, however, higher in MCs than TCs (*Figure 6—figure supplement 2D*), consistent with stronger lateral and feedforward inhibition among MCs than TCs (*Christie et al., 2001*; *Geramita and Urban, 2016*; *Geramita et al., 2016*; *Geramita and Urban, 2017*). Importantly, however, the high rates of evoked IPSCs observed in both cell types support the possibility that synaptic inhibition gates both TC and MC firing to drive gamma-frequency spike-time synchrony.

Within the barrage of synaptic inhibition, many IPSCs were indeed synchronized in both MC pairs and TC pairs (*Figure 6A, D, G and J*), and cross-correlograms of IPSC times revealed prominent central peaks in both cell types (*Figure 6B, E, H and K*). Surprisingly, despite the marked differences in spike-time synchrony, there was no cell-type difference in central peak heights of IPSC-time cross-correlograms (*Figure 6M*). Multiglomerular activation thus does not evoke more synchronous inhibition among TCs than MCs.

While this finding does not support our first hypothesis, synchronous synaptic inhibition may still drive greater synchronization of TC than MC firing in ways not readily apparent from the cross-correlogram analysis. In particular, cell-type differences in the (1) relative rates or amplitudes of synchronous vs. asynchronous IPSCs, (2) temporal distribution of IPSC synchrony throughout the photostimulation protocol, or (3) precision of IPSC synchrony may all generate differences in spike-time synchrony within the temporal gating framework. We therefore examined each possibility in turn.

Decomposition of inhibitory input into synchronous ($|\Delta t_{IPSC}| \leq 1$ ms) and asynchronous ($|\Delta t_{IPSC}| > 1$ ms) IPSCs enabled comparison of their relative rates, amplitudes, and temporal distributions across MCs and TCs (*Figure 6—figure supplement 3A–H*). In both cell types, synchronous IPSCs appeared tonically throughout the photostimulation protocol at rates greater than that observed spontaneously, but constituted the minority of IPSCs. Indeed, across all cells, the ratio of synchronous to asynchronous IPSC rates increased upon photostimulation onset to a level consistently less than 1 (*Figure 6—figure supplement 3I and J*). In contrast, photostimulation evoked synchronous IPSCs with amplitudes comparable to or even larger than asynchronous IPSCs, yielding a constant ratio of synchronous to asynchronous IPSC amplitudes slightly above 1 (*Figure 6—figure supplement 3K and L*). Critically, however, neither the relative rate nor amplitude of synchronous to asynchronous IPSCs was greater among TCs than MCs. Moreover, varying the time window within which evoked IPSCs were classified as synchronous ($w_{IPSC}$), thus providing a measure of synchrony precision (*Schoppa, 2006*), yielded a consistently larger fraction of synchronous IPSCs ($F_{synch}$) among MCs than TCs (*Figure 6O*). Our results thus firmly establish that the greater synchronization of TC than MC firing does not emerge from more synchronous synaptic inhibition among TCs than MCs, refuting our first hypothesis.

To evaluate whether distinct synchronization of synaptic inhibition across fast- and slow-gamma frequencies parallels the distinct spectral patterns of synchrony among TC and MC firing, we applied an identical CPSD analysis as above to the recorded IPSC times. While multiglomerular activation indeed evoked a visible increase in IPSC-time synchrony across fast frequencies in both cell types (*Figure 6C, F, I and L*), this increase was fundamentally distinct from the increase observed in spike-time synchrony. In particular, periodic synchronization of IPSC times was weaker, spread across a

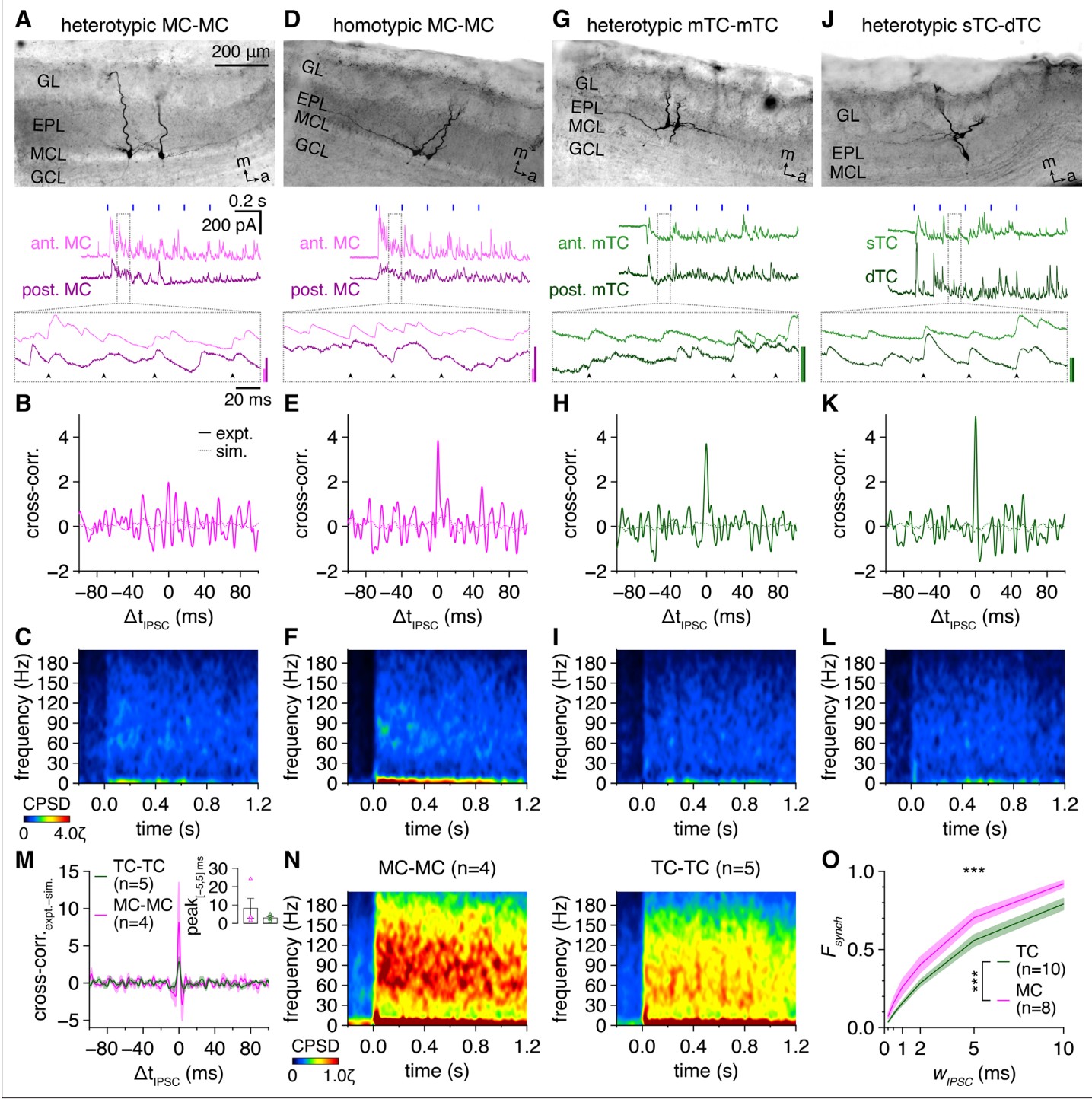

**Figure 6.** Multiglomerular activation evokes weak synchronization of inhibitory postsynaptic current (IPSC) times across gamma frequencies among both mitral cells (MCs) and tufted cells (TCs). (**A**) Example voltage-clamp recording of a heterotypic MC pair during photostimulation. Morphology (upper) and representative trial (lower; blue rectangles: 10 ms light pulses of the 5 Hz photostimulation protocol; arrowheads: synchronous IPSCs, $|\Delta t_{IPSC}| \leq 1$ ms). Inset scale bars: 100 pA. (**B**) Trial-averaged cross-correlogram of IPSC times recorded throughout the photostimulation protocol in the MC pair in (**A**) ('expt.') compared to the cross-correlogram of IPSC times simulated from rate-matched independent Poisson processes ('sim.'). (**C**) Trial-averaged IPSC-time cross-power spectral density (CPSD) spectrogram from the MC pair in (**A**) following photostimulation onset at 0.0 s. Color is scaled by multiples of the 95th percentile of 40–200 Hz CPSD values observed throughout the photostimulation protocol in all pairs ($\zeta$). No ridge analysis was applied given comparable levels of experimental and simulated IPSC-time CPSD across gamma frequencies (*Figure 6—figure supplement 4*). (**D–L**) Same as (**A–C**) for a homotypic MC pair (**D–F**), a heterotypic mTC pair (**G–I**), and a heterotypic sTC-dTC pair (**J–L**). Scaling in (**F, I ,L**) is the same as in (**C**). (**M**) Mean

*Figure 6 continued on next page*

*Figure 6 continued*

IPSC-time cross-correlograms (following subtraction of simulated IPSC-time cross-correlograms) of TC pairs and MC pairs. Inset: cross-correlogram peaks within $|\Delta t_{IPSC}| \leq 5$ ms tended to be higher across MC than TC pairs (two-sample t-test: p=0.30, $t_7 = 1.1$). (**N**) IPSC-time CPSD spectrograms averaged across all MC pairs (left) and TC pairs (right). (**O**) MCs exhibited a consistently greater fraction of IPSCs occurring synchronously ('$F_{synch}$') within a specific time window ('$w_{IPSC}$') than TCs (two-way ANOVA, cell type × $w_{IPSC}$; significant main effect of cell type, p=1.7 × 10⁻⁷, $F_{1,96} = 31.9$; significant main effect of $w_{IPSC}$, p=5.9 × 10⁻⁵⁰, $F_{5,96} = 208.9$; no significant interaction, p=0.50, $F_{5,96} = 0.87$).

The online version of this article includes the following figure supplement(s) for figure 6:

**Figure supplement 1.** Multiglomerular activation evokes a tonic increase in inhibitory postsynaptic current (IPSC) rates among both mitral cells (MCs) and tufted cells (TCs).

**Figure supplement 2.** Spontaneous and evoked inhibitory postsynaptic current (IPSC) properties are similar among mitral cells (MCs) and tufted cells (TCs).

**Figure supplement 3.** Mitral cells (MCs) and tufted cells (TCs) exhibit comparable proportions of evoked synchronous to asynchronous inhibitory postsynaptic current (IPSC) rates and amplitudes.

**Figure supplement 4.** Mitral cell (MC) pairs and tufted cell (TC) pairs do not exhibit greater gamma-frequency inhibitory postsynaptic current (IPSC)-time synchrony than IPSC times simulated from rate-matched independent Poisson processes.

**Figure supplement 5.** Tufted cells (TCs) do not exhibit greater gamma-frequency inhibitory postsynaptic current (IPSC)-time synchrony.

**Figure supplement 6.** Inhibitory postsynaptic current (IPSC)-time synchrony does not correlate with within-pair intersomatic distance.

**Figure supplement 7.** Tufted cell (TC) inhibitory postsynaptic current (IPSC)-time synchrony does not correlate with within-pair intersomatic differences in external plexiform layer (EPL) depth.

**Figure supplement 8.** Gamma-frequency synchronization of inhibitory currents is weak among both mitral cells (MCs) and tufted cells (TCs).

broader frequency range, and less concentrated into distinct epochs than periodic spike-time synchrony. CPSD analysis of simulated IPSC times generated from independent Poisson processes (*Figure 6—figure supplement 1*) in fact revealed that the modest increase in periodic synchronization of IPSC times in both cell types could be attributed entirely to an increase in chance levels of synchrony with increasing IPSC rates (*Figure 6—figure supplement 4*). Both time-dependent (*Figure 6N*) and time-independent CPSD analyses (*Figure 6—figure supplement 5*) across all pairs supported this conclusion, with modestly higher IPSC-time periodic synchrony among MCs than TCs paralleling the modestly higher evoked IPSC rates among MCs than TCs. These results were further unrelated to any cell-type differences in the proportion of homotypic-to-heterotypic pairs (chi-squared test: p=0.073, $\chi^2 = 3.2$) or intersomatic position (*Figure 6—figure supplement 6*, *Figure 6—figure supplement 7*). Cell-type differences in gamma-frequency synchronization of synaptic inhibition thus do not account for the dynamic synchronization of TC firing across fast- and slow-gamma frequencies and the more stable synchronization of MC firing at slow-gamma frequencies, refuting our second hypothesis.

As a caveat, the above analyses rely on accurate IPSC detection, which can be difficult during barrages of input. However, equivalent cross-correlogram and CPSD analyses of the raw inhibitory currents recorded throughout the photostimulation protocol likewise failed to reveal more synchronous inhibition among TCs than MCs (*Figure 6—figure supplement 8*).

In summary, our results thus show that differences in synaptic inhibition among TCs and MCs do not underlie the cell-type differences in gamma-frequency spike-time synchrony. Together with the lack of gamma-frequency synchrony in excitatory inputs (*Figure 2—figure supplement 1F*), the minimal gamma-frequency synchrony in inhibitory inputs stands in stark contrast to the precise gamma-frequency synchronization of synaptic excitation and inhibition observed during fast network oscillations driven by temporal gating elsewhere in the brain (e.g., *Whittington et al., 1995*; *Fisahn et al., 1998*; *Hasenstaub et al., 2005*; *Atallah and Scanziani, 2009*). Consequently, while synaptic inhibition remains a necessary component of fast network oscillations in the MOB, some other cellular or circuit feature must account for the profound differences observed in periodic synchronization of TC and MC firing, motivating consideration of alternative mechanisms of gamma-frequency synchrony.

## Greater oscillatory behavior among resonant TCs than MCs supports dense gamma-frequency spike-time synchrony

Previously, we used step-current injections overlaid with simulated synaptic currents to demonstrate that aperiodic synaptic input can shift or 'reset' the phase of roughly periodically-firing MCs to produce fast-timescale, dense periodic spike-time synchrony (*Galán et al., 2006*). Whether such

a phase-resetting mechanism contributes to sensory-evoked fast network oscillations in the MOB remains untested, but provides an attractive alternative mechanism whereby cell-type differences in firing periodicity (i.e., oscillatory behavior) and/or phase-resetting properties may act on comparable synaptic input to produce distinct patterns of periodic spike-time synchrony. In accordance with a phase-resetting mechanism, we therefore hypothesized that (1) oscillatory behavior among individual cells correlates with spike-time synchrony within pairs, with TCs exhibiting greater gamma-frequency firing periodicity than MCs; and (2) periodic spike-time synchrony is dense, with firing synchronized across consecutive fast- and slow-gamma-frequency cycles among TCs.

To analyze oscillatory behavior, we returned to our original dataset and first examined spike-time auto-power spectral density (APSD; i.e., the power spectrum of the auto-correlogram; *Figure 7A–D*). Both MCs and TCs exhibited more periodic gamma-frequency firing than rate-matched Poisson processes (*Figure 7—figure supplement 1*), with cell-type differences in APSD spectrograms closely matching the cell-type differences in CPSD spectrograms among pairs. Specifically, TCs exhibited highly periodic firing across both fast- and slow-gamma frequencies, while MCs exhibited less periodic firing across predominantly slow-gamma frequencies (*Figure 7F*). Indeed, maximal ridge detection using a threshold ($\lambda$) equal to the 95th percentile of 40–200 Hz APSD values across all cells confirmed that 85% of TCs exhibited ≥1 APSD ridge compared to only 44% of MCs (*Figure 7E*), with epochs of robust periodic firing among TCs extending across higher frequencies and APSD levels than MCs (*Figure 7G–I*, *Figure 7—figure supplement 2A*). Examination of spike-time APSD averaged throughout the photostimulation protocol and independent of ridge detection directly reinforced these findings (*Figure 7—figure supplement 2B and C*). Greater TC than MC oscillatory behavior further persisted throughout the average photostimulation cycle (*Figure 7—figure supplement 3*).

As a complementary analysis of oscillatory behavior, we also calculated CV2 (i.e., the normalized variance across consecutive interspike intervals [ISIs]) (*Holt et al., 1996*). Within each photostimulation cycle, MCs frequently exhibited widely distributed CV2, consistent with broad ISI distributions and modest periodicity. In contrast, TCs exhibited narrower ISI distributions and CV2 clustered near 0 (*Figure 7—figure supplement 2D–G*). Indeed, across all cells, median CV2 was significantly lower among TCs than MCs (*Figure 7—figure supplement 2H*). Thus, despite the more phasic firing pattern of TCs than MCs, TC firing remained more periodic from moment to moment than MC firing. Together with the above spectral analysis, these results confirm that TCs exhibit greater gamma-frequency firing periodicity than MCs, consistent with our first hypothesis.

To evaluate whether such oscillatory behavior indeed promotes spike-time synchrony in the MOB, we first classified pairs in which both cells exhibited ≥1 APSD ridge as periodic, and the remaining pairs as aperiodic. Strikingly, epochs of robust periodic spike-time synchrony, manifest in spike-time CPSD ridges, were almost exclusively detected among periodic pairs, independent of cell type (*Figure 7J*). Spike-time synchrony independent of periodicity, as measured by cross-correlogram central peaks, was likewise markedly greater among periodic than aperiodic pairs (*Figure 7K*). On a per-pair basis, spike-time cross-correlogram central peaks further directly correlated with spike-time APSD averaged throughout the photostimulation protocol in both MCs and TCs considered together or separately (*Figure 7L–N*). Oscillatory behavior thus promotes spike-time synchrony in the MOB, with TCs exhibiting greater gamma-frequency firing periodicity than MCs, confirming our first hypothesis.

Fast network oscillations in hippocampal and neocortical networks emerge from the sparse synchronization of principal cells firing at irregular rates well below gamma frequencies (*Wang, 2010*). In contrast to these regions, the instantaneous firing rates of MCs and TCs specifically during epochs of robust periodic synchrony (i.e., spike-time CPSD ridges) instead closely matched instantaneous CPSD ridge frequencies (*Figure 8A–E*), with deviations largely limited to abrupt transitions in firing rate (e.g., immediately following photostimulation) that exceeded the finite resolution of our spectral analysis. Indeed, across all cells, the mean correspondence between instantaneous firing rate and the frequency of periodic spike-time synchrony approached unity in both cell types (*Figure 8F*), with TC firing extending into faster frequencies than MC firing to match the fast-gamma-frequency synchrony widely observed among TCs (*Figure 8G*). Such close correspondence between firing rate and periodic synchrony – indicative of spike-time synchrony across multiple consecutive oscillatory cycles and particularly evident among TCs (*Figure 1G and J*) – directly agrees with dense spike-time synchrony arising from the phase-resetting of periodically firing neurons. Further supporting this conclusion, relative firing rate differences, which can attenuate phase-resetting-mediated synchronization (***Burton***

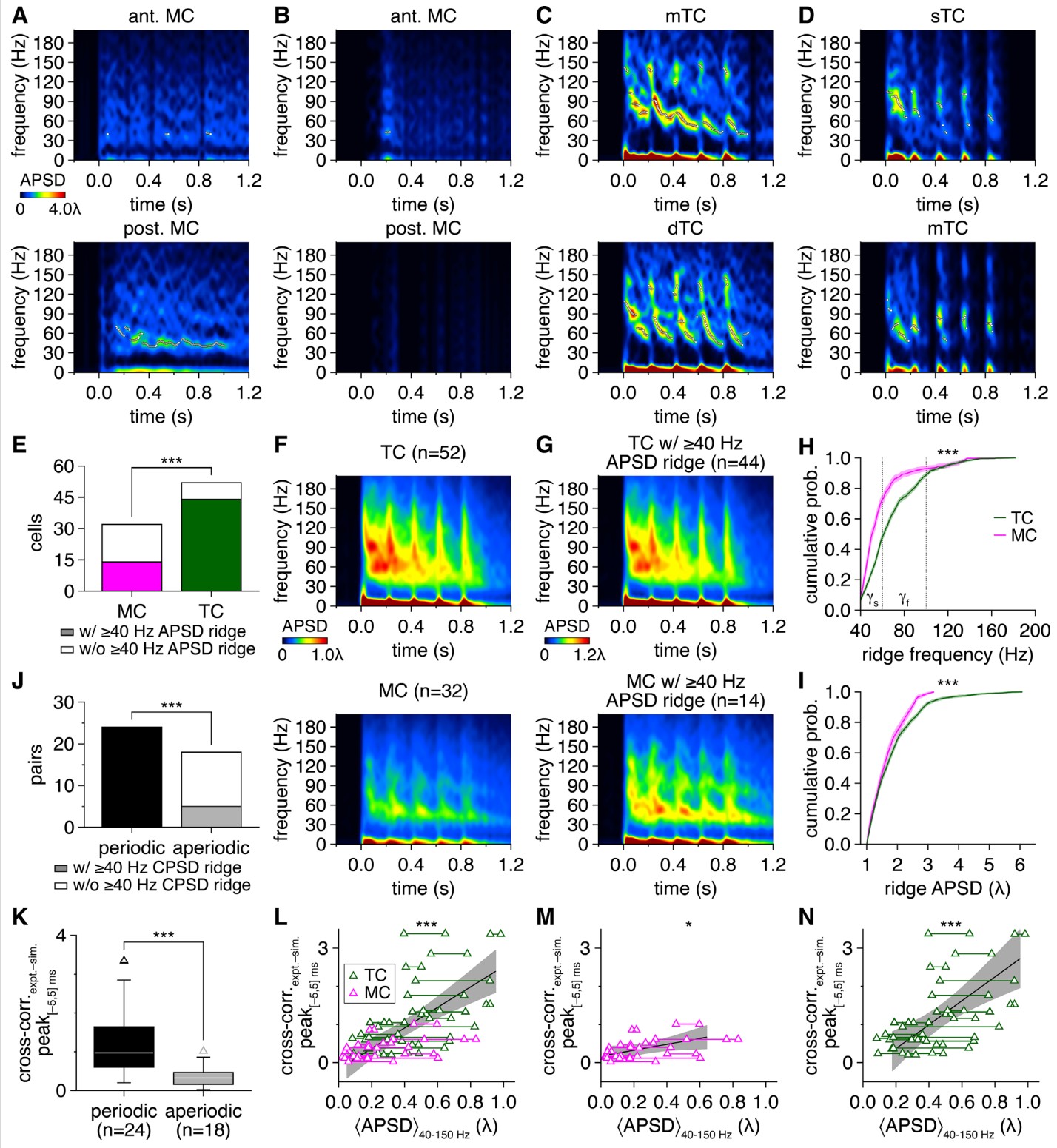

**Figure 7.** Greater oscillatory behavior among tufted cells (TCs) than mitral cells (MCs) promotes gamma-frequency spike-time synchrony. (**A**) Trial-averaged spike-time auto-power spectral density (APSD) spectrogram from the MCs in *Figure 1A*. Continuous epochs ($\Delta$Hz/ms < 150) of high APSD reflecting robust periodic firing are defined as ridges and demarcated with white circles. Color is scaled by multiples of the ridge threshold ($\lambda$). (**B–D**) Same as (**A**) for the MCs and TCs in *Figure 1D, G and J*. (**E**) More TCs than MCs exhibited spike-time APSD ridges (chi-squared test: p=8.3 × 10^{-5}, $\chi^2$ = 15.5). (**F**) Spike-time APSD spectrograms averaged across all TCs (upper) and MCs (lower). (**G**) Spike-time APSD spectrograms averaged across

*Figure 7 continued on next page*

*Figure 7 continued*

all TCs (upper) and MCs (lower) exhibiting APSD ridges. (**H, I**) Cumulative distributions of frequencies (**H**) and APSD (**I**) across all spike-time APSD ridges. TCs exhibited faster (**H**) (two-sample Kolmogorov–Smirnov test: p=2.9 × 10$^{-49}$) and more precise (**N**) (two-sample Kolmogorov–Smirnov test: p=2.5 × 10$^{-9}$) gamma-frequency periodic firing than MCs. Shading denotes 95% confidence intervals. (**J**) Periodically firing pairs (comprised of cells with ≥1 APSD ridge) were substantially more likely to exhibit periodic spike-time synchrony (i.e., ≥ 1 spike-time cross-power spectral density [CPSD] ridge) than aperiodically firing pairs (chi-squared test: p=5.4 × 10$^{-7}$, $\chi^2$ = 25.1). (**K**) Spike-time cross-correlogram peaks within |Δt$_{spike}$| ≤ 5 ms were higher among periodically firing than aperiodically firing pairs (Wilcoxon rank-sum test: p=7.0 × 10$^{-5}$). (**L–N**) Periodicity in firing (i.e., spike-time APSD averaged across 40–150 Hz and throughout the photostimulation protocol), averaged across cells of each pair, positively correlated with spike-time synchrony independent of periodicity (i.e., spike-time cross-correlogram central peak heights) among all MC pairs and TC pairs combined (**L**) (linear regression, slope significantly different from 0: p=6.4 × 10$^{-7}$, t$_{40}$ = 5.9; R$^2$ = 0.47), among MC pairs alone (**M**) (linear regression, slope significantly different from 0: p=0.040, t$_{14}$ = 2.3; R$^2$ = 0.27), and among TC pairs alone (**N**) (linear regression, slope significantly different from 0: p=1.7 × 10$^{-4}$, t$_{24}$ = 4.5; R$^2$ = 0.45). Shading denotes 95% confidence interval.

The online version of this article includes the following figure supplement(s) for figure 7:

**Figure supplement 1.** Mitral cells (MCs) and tufted cells (TCs) exhibit greater gamma-frequency spike-time periodicity than spike times simulated from rate-matched independent Poisson processes.

**Figure supplement 2.** Tufted cells (TCs) exhibit greater oscillatory behavior than mitral cells (MCs) on both a per-cell basis and independent of ridge-based or spectral analyses.

**Figure supplement 3.** Greater periodicity of tufted cell (TC) than mitral cell (MC) firing persists throughout the average 5 Hz photostimulation cycle.

*et al., 2012*; *Zhou et al., 2013*), were both lower within TC than MC pairs and, independent of cell type, negatively correlated with cross-correlogram and CPSD measures of spike-time synchrony (*Figure 8—figure supplement 1*).

That TCs exhibit greater oscillatory behavior than MCs despite comparable synaptic input following multiglomerular activation suggests that the intrinsic biophysical properties of TCs yield greater tendency toward oscillatory behavior (i.e., resonance) than the biophysical properties of MCs. Therefore, to begin to trace the cell-type differences in periodic spike-time synchrony to potential biophysical sources, in our final set of analyses we examined subthreshold oscillations (STOs) – a manifestation of intrinsic resonance (*Hutcheon and Yarom, 2000*) – among MCs and TCs. As our cell-attached recordings did not permit isolation of STOs, we instead re-examined a previously collected in vitro dataset of MC and TC step-current responses in the presence of synaptic antagonists (*Burton and Urban, 2014*). Within this dataset, some MCs fired one-to-a-few spike clusters interspersed with multiple putative STOs per step-current injection, with putative STOs generating local maxima within continuous Morlet wavelet transform spectrograms of subthreshold membrane potentials (*Figure 9A–C*). Other MCs, in contrast, exhibited adapting firing patterns with few putative STOs evident and spectrograms dominated by residual low-frequency components of interpolated spikes (*Figure 9F–H*). To quantitatively assess resonance across cells, STOs were isolated via iterative semi-automated ridge detection and post-hoc visual confirmation (see Materials and methods), as performed elsewhere (*Fourcaud-Trocmé et al., 2018*). Confirming our visual inspection, a natural division in MCs emerged from this analysis, with half of MCs exhibiting multiple STOs per step current and the other half exhibiting few or no STOs per step current. Operationalizing this division, we classified cells as resonant if they exhibited ≥10 STO events in total, corresponding to the 10 step currents assayed (50–500 pA, in steps of 50 pA; one trial per step).

As previously noted (*Burton and Urban, 2014*), TCs were more likely to respond to depolarization with clusters of periodic high-frequency spikes than MCs. Inspection of the subthreshold epochs interposing those clusters revealed numerous STOs per TC response (*Figure 9K, L, P and Q*). Applying our operational metric, 92% of TCs proved resonant compared to only 50% of MCs (*Figure 9U*). Subthreshold resonance is thus more widespread among TCs than MCs.

To assess how subthreshold resonance translates into oscillatory behavior, we examined the relationship between spikes and immediately preceding STOs (separated by ≤2 STO periods), using ridge maxima (*Figure 9C, M and R*) and least squares estimate-sinusoid fits (*Figure 9B, L and Q*) to extract STO period and phase, respectively. Among resonant TCs and MCs, nearly all cells exhibited robust phase-locking of spike times to the preceding STO (*Figure 9V*). MC spikes consistently occurred at phases just prior to the STO peak (*Figure 9D*), matching previous findings (*Desmaisons et al., 1999*). TC spike times exhibited identical phase-locking (*Figure 9N and S*), with no difference in mean spike phase observed between cell types (*Figure 9V*). Following the

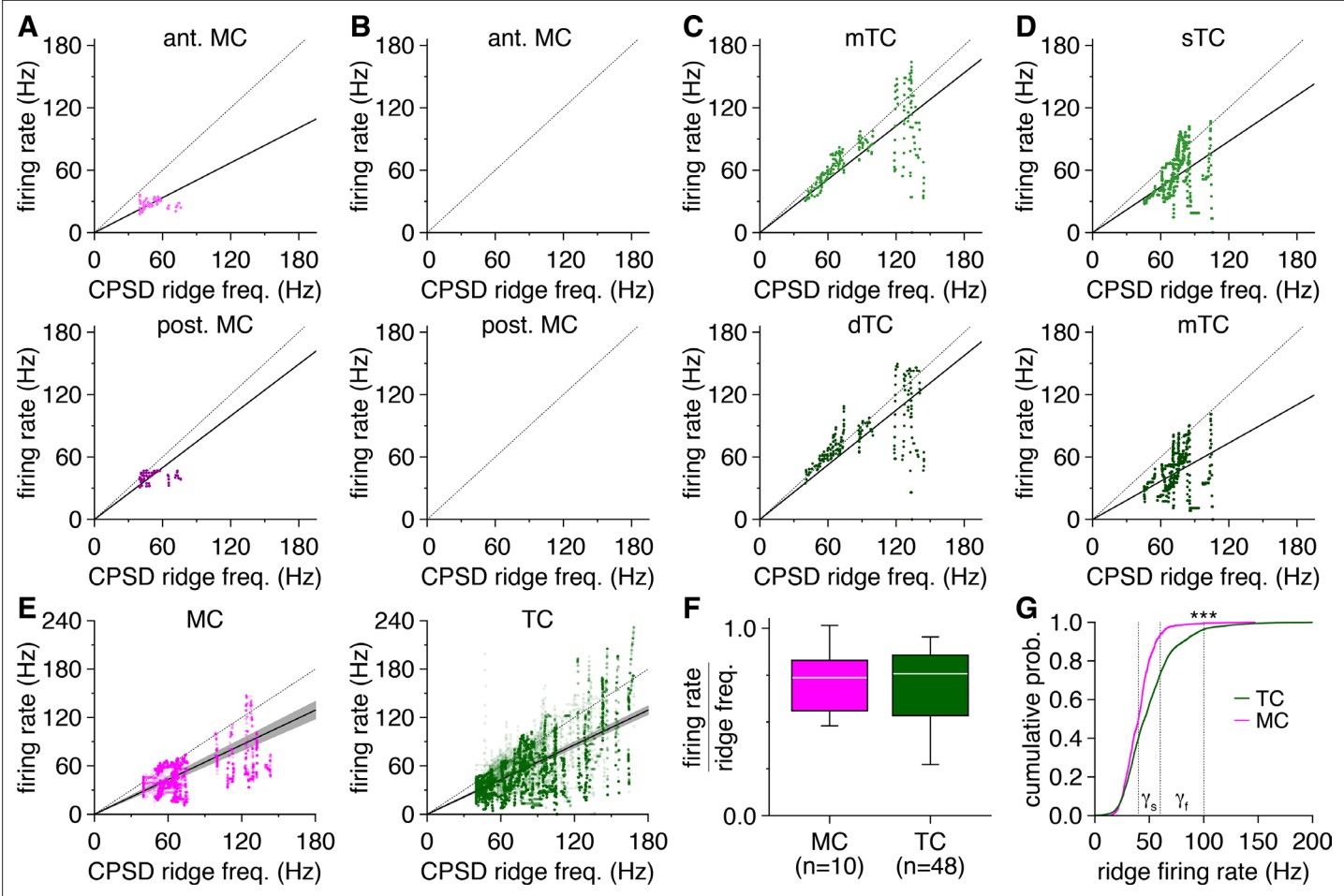

**Figure 8.** Mitral cell (MC) pairs and tufted cell (TC) pairs exhibit gamma-frequency spike-time synchrony specifically when firing at gamma frequencies. (**A**) Instantaneous firing rate of the MCs in *Figure 1A* plotted against the simultaneous frequency of periodic spike-time synchrony during each spike-time cross-power spectral density (CPSD) ridge. Dashed line: unity. Solid line: mean firing rate to CPSD ridge frequency ratio. (**B–D**) Same as (**A**) for the MCs and TCs in *Figure 1D, G and J*. (**E**) Instantaneous firing rate plotted against the simultaneous frequency of periodic spike-time synchrony across all MCs (left) and TCs (right). Darker coloring denotes overlapping data points. Solid lines: mean firing rate to CPSD ridge frequency ratio, averaged across cells. (**F**) Instantaneous firing rates relative to the simultaneous frequency of periodic spike-time synchrony were comparable among MCs and TCs (Wilcoxon rank-sum test: p=0.89) and approached unity in both cells, consistent with synchronization of periodic firing across the majority of spikes fired during an epoch. (**G**) Cumulative distribution of instantaneous TC and MC firing rates recorded during spike-time CPSD ridges. TCs exhibiting gamma-frequency spike-time synchrony fired at higher rates than MCs exhibiting gamma-frequency spike-time synchrony (two-sample Kolmogorov–Smirnov test: p=0).

The online version of this article includes the following figure supplement(s) for figure 8:

**Figure supplement 1.** Firing rate differences attenuate periodic spike-time synchrony.

initial post-STO spike, however, TC firing persisted at instantaneous rates closely matching STO frequencies (*Figure 9L, O, Q and T*), while MC firing was visibly outpaced by the preceding STO (*Figure 9B and E*), a difference particularly evident at the population level (*Figure 9W*). Indeed, across all resonant cells, TCs exhibited a 1:1 relationship between firing rate and STO frequency while MCs exhibited a 1:2 relationship (*Figure 9W*). Subthreshold resonance is thus not only more widespread among TCs than MCs, but it also more faithfully entrains TC than MC firing. Critically, these results not only implicate specific conductances involved in STO generation with the greater oscillatory behavior among TCs than MCs (see Discussion), but additionally identify synchronization of STOs as a possible mechanism sustaining dense gamma-frequency spike-time synchrony across gaps in TC firing (e.g., *Figure 1G and J*).

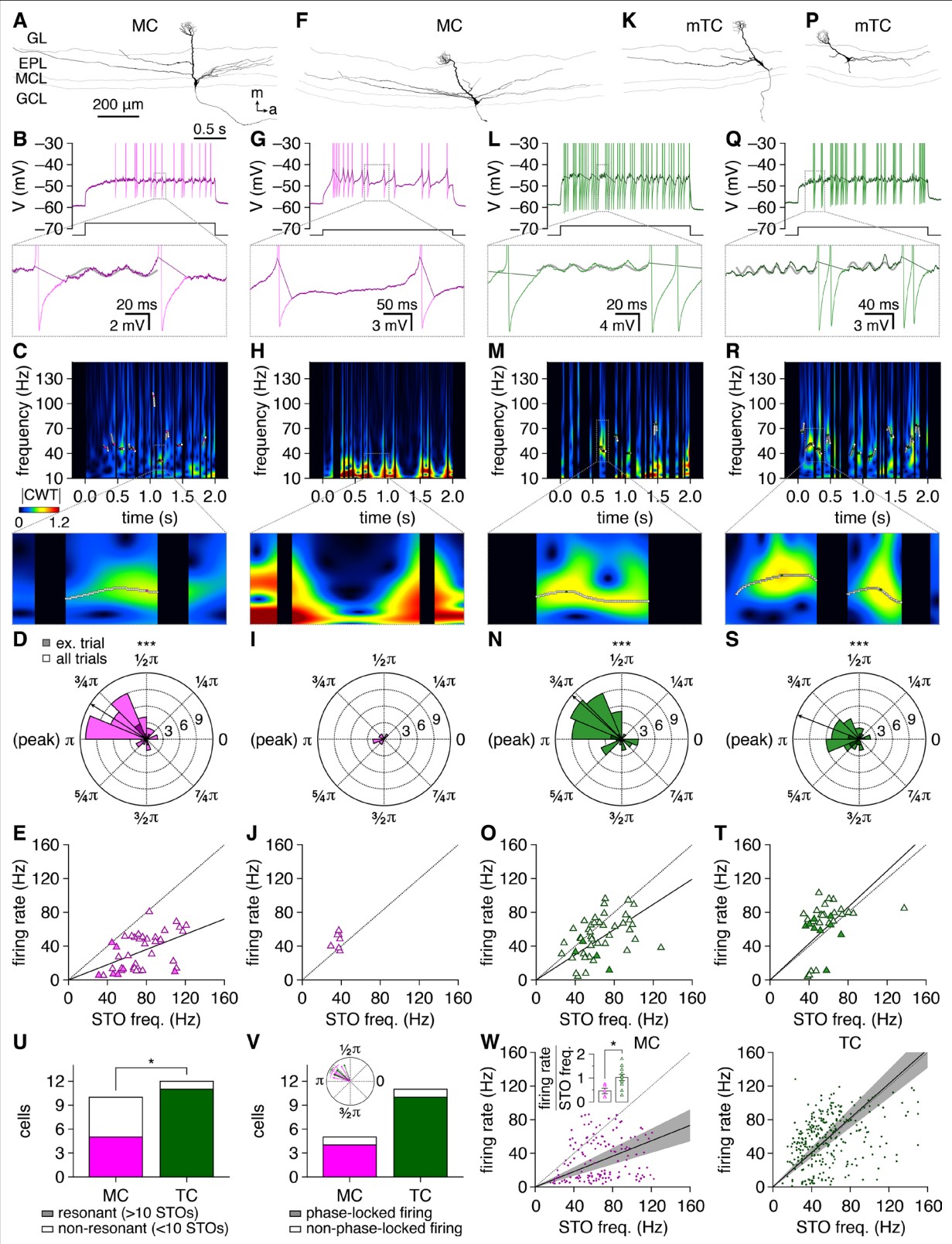

**Figure 9.** Intrinsic resonance is more widespread and better entrains firing among tufted cells (TCs) than mitral cells (MCs). (**A, B**) Example recording of a resonant MC. Reconstructed morphology (**A**) and representative response to depolarizing step current injection (**B**) (light trace: original membrane potential; dark trace: subthreshold membrane potential; gray line: least squares estimate-sinusoid fit to subthreshold oscillation [STO]). (**C**) Spectrogram showing continuous Morlet wavelet transform (CWT) of subthreshold membrane potential from the MC response in (**B**). CWT ridges are confirmed as

*Figure 9 continued on next page*

*Figure 9 continued*

STOs following post-hoc visual inspection of the underlying membrane potential and demarcated with white circles. Colored circles: ridge maxima, defining frequency of confirmed STO. (**D**) Post-STO spike phases for the MC in (**A**), showing a non-uniform distribution at phases just prior to the STO peak (Rayleigh's test: $p_{BH}$ = 2.0 × 10$^{-8}$). Dark-colored bars: spike phases from the example response in (**B**). Arrow: median spike phase. (**E**) Instantaneous firing rates were consistently slower than the preceding STO frequency in the MC in (**A**). Dashed line: unity. Solid line: mean firing rate to STO frequency ratio. Filled symbols: STOs detected in the example response in (**B**). (**F–J**) Same as (**A–E**) for a non-resonant MC. (**K–O**) Same as (**A–E**) for a resonant mTC. Post-STO spike phases were non-uniformly distributed at phases just prior to the STO peak (**N**) (Rayleigh's test: $p_{BH}$ = 1.0 × 10$^{-6}$). (**P–T**) Same as (**A–E**) for a second resonant mTC. Post-STO spike phases were non-uniformly distributed at phases just prior to the STO peak (**S**) (Rayleigh's test: $p_{BH}$ = 3.2 × 10$^{-5}$). (**U**) More TCs than MCs were resonant (chi-squared test: p=0.029, $\chi^2$ = 4.8). (**V**) Post-STO spikes were significantly phase-locked (i.e., non-uniformly distributed) among comparable proportions of resonant MCs and TCs (chi-squared test: p=0.54, $\chi^2$ = 0.37), encompassing the vast majority of resonant cells. Inset: resonant MCs and TCs with phase-locked firing exhibited comparable median spike phases (Watson–Williams test: p=0.87). (**W**) Instantaneous firing rate vs. preceding STO frequency across all resonant MCs (left) and TCs (right). Solid lines: mean firing rate to STO frequency ratio, averaged across cells. Inset: TCs exhibited closer entrainment of firing rate to preceding STO frequency than MCs (two-sample t-test: p=0.020, $t_{14}$ = 2.6).

## Discussion

Identifying the mechanisms underlying gamma-frequency oscillations in the MOB will be key to understanding how fast network oscillations contribute to olfactory coding and behavior. Here, we uncovered profound cell-type differences in gamma-frequency spike-time synchrony among principal MCs and TCs. Specifically, multiglomerular activation evoked more widespread and precise periodic synchronization of TC than MC firing that persisted throughout the theta-frequency sensory-input cycle. TC synchrony further frequently extended across fast-gamma frequencies with a sweeping deceleration toward slow-gamma frequencies – directly mirroring MOB LFP recordings in vivo – while MC synchrony was concentrated in slow-gamma frequencies. Mechanistically, greater synchronization arose among cells with convergent rather than divergent apical dendrites but occurred independent of intraglomerular lateral excitation, which was selectively absent among TCs. Surprisingly, cell-type differences in periodic spike-time synchrony could likewise not be traced to any discernable difference in the synchronization of synaptic inhibition, in contrast with temporal gating mechanisms of fast network oscillations elsewhere in the brain. Instead, greater TC than MC spike-time synchrony directly paralleled the greater resonant oscillatory behavior among TCs than MCs and emerged in patterns consistent with a densely synchronous network oscillation. Collectively, our results thus argue that synchronization of periodically firing TCs, likely mediated by a phase-resetting mechanism, strongly contributes to fast network oscillations in the MOB.

### Fast- and slow-gamma-frequency synchrony in the MOB

For decades, the MOB has served as a prominent model circuit for investigating fast network oscillations (*Rojas-Líbano and Kay, 2008*). Sensory-evoked gamma-frequency oscillations in particular are generated intrinsically within the MOB (*Gray and Skinner, 1988*; *Neville and Haberly, 2003*; *Martin et al., 2004*; *Martin et al., 2006*) and have been extensively studied in vitro by electrically stimulating OSNs in acute MOB slices. However, while fast- and slow-gamma-frequency oscillations have been widely observed in MOB LFP recordings in vivo (*Kay, 2003*; *Lepousez and Lledo, 2013*; *Manabe and Mori, 2013*; *Frederick et al., 2016*; *Zhuang et al., 2019*), MC synchrony and network oscillations recorded in vitro have been exclusively confined to slow-gamma frequencies (and lower), with peak periodicity observed at <30 Hz (*Friedman and Strowbridge, 2003*), ~40–50 Hz (*Lagier et al., 2004*; *Schoppa, 2006*; *Gire and Schoppa, 2008*; *Pandipati et al., 2010*; *Pandipati and Schoppa, 2012*; *Pouille et al., 2017*), and ~55–65 Hz (*Bathellier et al., 2006*; *Lagier et al., 2007*). Without exception, however, recordings in each of these in vitro studies specifically targeted MCs or the MC layer, and spectral analyses were frequently averaged over a broad post-stimulation window, often excluding the initial ~30–100 ms to avoid stimulus artifacts. By targeting our recordings to both principal cell types, employing an optogenetic protocol with negligible stimulus artifacts, and performing complementary static and dynamic spectral analyses, we discovered robust periodic spike-time synchrony in acute slices demonstrably matching the frequency, dynamics, and even theta-frequency coupling of fast- and slow-gamma-frequency oscillations in vivo. Beyond identifying spike-time synchrony among TCs as a major component of gamma-frequency oscillations in the MOB, our results thus also reaffirm the outstanding facility of the acute slice preparation for mechanistic investigations of fast network oscillations in the MOB.

Differential spike-time synchrony among TCs and MCs, the two types of excitatory projection neurons in the MOB, will critically influence how information is transmitted to downstream brain regions. Exploring the impact that TC synchrony in particular has on synaptic communication with two major downstream targets – anterior piriform cortex and olfactory tubercle, regions with prominent roles in olfactory consciousness and hedonic processing (*Wesson and Wilson, 2011*; *Mori et al., 2013*) – stands as a promising direction for future research, especially given that the cross-frequency coupling of fast-gamma-frequency TC synchrony to theta-frequency sensory input cycles effectively recapitulates classic theta-burst protocols for inducing robust long-term potentiation (*Colgin, 2015*).

Fast- and slow-gamma-frequency oscillations in the MOB share features with fast- and slow-gamma-frequency oscillations elsewhere in the brain, suggesting potential similarity in broad functional principles, if not precise mechanisms. In particular, similar to the nesting of fast- and slow-gamma-frequency oscillations within theta-frequency sensory-input cycles in the MOB (*Lepousez and Lledo, 2013*; *Manabe and Mori, 2013*; *Zhuang et al., 2019*), hippocampal CA1 exhibits prominent cross-frequency coupling of fast- and slow-gamma-frequency oscillations to an underlying theta-frequency oscillation critical in mnemonic processing (*Buzsáki and Wang, 2012*; *Colgin, 2015*). In CA1, however, fast- and slow-gamma-frequency synchronization of principal cells is driven extrinsically by shifting communication between medial entorhinal cortex and hippocampal CA3, respectively (*Colgin et al., 2009*), while our results in the MOB instead point toward differential synchronization of complementary cell types receiving common inputs. These differences notwithstanding, leading hypotheses respectively associate fast- vs. slow-gamma-frequency oscillations in CA1 with the encoding of current spatial information vs. spatial memory retrieval (*Colgin, 2015*), functions provocatively similar to burgeoning evidence respectively linking TC vs. MC activity to the encoding of current olfactory surroundings vs. learned olfactory context (*Burton et al., 2020*). This potential functional correspondence between fast- and slow-gamma-frequency oscillations of the MOB and hippocampus, while speculative, warrants further investigation.

## Mechanisms and functions of intraglomerular enhancement of spike-time synchrony

Previous investigation of sensory-evoked spike-time synchrony between pairs of MOB principal cells has focused exclusively on heterotypic MCs (*Kashiwadani et al., 1999*; *Schoppa, 2006*). Whether convergence of apical dendrites within the same glomerulus – engaging shared intraglomerular circuits and sensory input – enhances or attenuates sensory-evoked spike-time synchrony was thus previously unknown. Here, we have confirmed that multiglomerular activation evokes greater periodic spike-time synchrony among homotypic than heterotypic principal cells, a result with several critical implications for sensory processing in the MOB. In particular, greater synchronization of homotypic than heterotypic pairs suggests that periodic spike-time synchrony may be more important to faithfully communicating the activation of a specific odorant receptor than in binding disparate elements of a sensory input into a single percept. Moreover, the likely short vs. long synaptic integration windows of EPL-interneuron vs. GC populations in the MOB suggests that more vs. less synchronous firing among homotypic vs. heterotypic principal cells, respectively, engages interneuronal circuits with distinct computational roles (*Burton, 2017*). Finally, developmental sensory experience may dramatically alter fast network oscillations in the MOB by specifically increasing the number of homotypic principal cells linked to the activated glomerulus (*Liu et al., 2016*), outlining a novel mechanism of experience-dependent temporal coding plasticity.

Further investigation is necessary to determine which intraglomerular circuit(s) promote gamma-frequency spike-time synchrony among homotypic principal cells. Complementary lines of evidence have established that lateral excitation and not electrical coupling synchronizes the irregular 0–10 Hz firing of homotypic MCs driven by step-current injection or bath NMDA application. In particular, AMPAR antagonists (but not NMDAR or GABAR antagonists) abolish spike-time synchrony without impacting electrical coupling (*Schoppa and Westbrook, 2002*). Moreover, lateral excitation amplitudes and asymmetry correlate with the strength and timing of spike-time synchrony (*Schoppa and Westbrook, 2002*) but do not correlate with the strength of electrical coupling (*Pimentel and Margrie, 2008*). However, both the limited precision of such aperiodic spike-time synchrony (–10 to +10 ms spike-lag) and prolonged kinetics of lateral excitation (12–23 ms EPSP half-width) (*Schoppa and Westbrook, 2002*; *Christie et al., 2005*) are not obviously compatible with the rapid timescale

of gamma-frequency synchrony. Likewise, inhibition and not excitation typically drives synchronization of periodically firing neurons (*Van Vreeswijk et al., 1994*; *Wang, 2010*), though this depends on the specific phase-resetting properties of neurons, which remain unknown for dendritic MC input. Finally, as we now demonstrate, homotypic TCs exhibit robust gamma-frequency spike-time synchrony without lateral excitation. This surprising result not only points toward principal cell electrical coupling as a more likely factor underpinning fast network oscillations in the MOB, consistent with results of connexin36 knockout (*Pouille et al., 2017*), but further reinforces the critical importance that subcellular positioning of gap junctions and presynaptic specializations has on neural communication within the glomerulus (*Bourne and Schoppa, 2017*).

How synchronization of TC firing influences the activity of MCs linked to the same glomerulus, or even whether multiglomerular activation can synchronize MC firing to TC firing, likewise remain open questions of outstanding interest. TCs can laterally excite MCs linked to the same glomerulus (*Pimentel and Margrie, 2008*; *Najac et al., 2011*), which not only underscores the surprising absence of lateral excitation between homotypic TCs, but also suggests that homotypic TC-MC pairs may parallel homotypic MC pairs in exhibiting synchronous irregular firing but more limited gamma-frequency synchrony. Consistent with this prediction, spontaneous firing within homotypic TC-MC pairs exhibits less precise spike-time synchrony than spontaneous firing within homotypic TC pairs (*Ma and Lowe, 2010*).

## Biophysical sources of intrinsic resonance and oscillatory behavior among MCs and TCs

Consistent with the contribution of phase-resetting to fast network oscillations in the MOB, intrinsic resonance supporting STOs has previously been observed in MCs both in vitro (*Chen and Shepherd, 1997*; *Desmaisons et al., 1999*; *Friedman and Strowbridge, 2000*; *Balu et al., 2004*; *Lagier et al., 2004*) and in vivo (*Debarbieux et al., 2003*; *Fourcaud-Trocmé et al., 2018*), can regulate MC spike timing and firing rate (*Desmaisons et al., 1999*) and phase-lock MC membrane potentials to gamma-frequency LFP oscillations (*Lagier et al., 2004*; *Fourcaud-Trocmé et al., 2018*), and supports gamma-frequency synchronization of MC firing in multiple biophysical models (*Brea et al., 2009*; *David et al., 2009*; *Li and Cleland, 2013*; *David et al., 2015*; *Li and Cleland, 2017*). Here, we now demonstrate that intrinsic resonance supporting STOs is both more widespread among TCs and more faithfully entrains TC than MC firing to gamma frequencies.

Similar to several other cell types with mixed-mode bursting or 'stuttering' firing patterns (*Wang, 1993*; *Gutfreund et al., 1995*; *Hutcheon and Yarom, 2000*), STOs in MCs emerge from the interplay between slow potassium currents, which confer both mixed-mode bursting and resonance (i.e., the amplification of select input frequencies), and a persistent sodium current, which amplifies resonance into detectable oscillations (*Desmaisons et al., 1999*; *Balu et al., 2004*; *Bathellier et al., 2006*; *Rubin and Cleland, 2006*). The greater propensity of TCs than MCs to exhibit mixed-mode bursting (*Burton and Urban, 2014*) and STOs suggests that TCs express homogenously high levels of slow potassium currents compared to more heterogenous expression among MCs (*Padmanabhan and Urban, 2014*). Functionally, such currents promote reliable encoding of theta-frequency-patterned inputs (*Balu et al., 2004*), and indeed, we observed higher levels of theta-frequency synchrony among TC than MC firing. Slow potassium currents and intrinsic resonance among TCs may thus be critically involved in communicating multiplexed theta- and gamma-frequency signals to downstream regions. Differential expression of slow potassium currents likely also influences the phase-resetting properties of TCs vs. MCs, identifying a key area for future investigation. Likewise, changes in resonance and/or phase-resetting properties by modulation of potassium currents (*Stiefel and Ermentrout, 2016*) may constitute a mechanism complementary to modulation of lateral inhibitory circuits (*Pandipati et al., 2010*; *Li and Cleland, 2013*; *Li et al., 2015*) for altering fast network oscillations in the MOB across behavioral states.

## Sparse vs. dense gamma-frequency synchronization of MOB principal cells

Lateral inhibitory circuits are critically involved in synchronizing principal cell firing to generate gamma-frequency oscillations in the MOB (*Lagier et al., 2004*; *Bathellier et al., 2006*; *Lagier et al., 2007*; *Lepousez and Lledo, 2013*; *Fukunaga et al., 2014*), though the precise mechanism driving synchrony

remains contested. Temporal gating of MC activity by synchronous GC-mediated inhibition, paralleling sparsely synchronous network oscillations elsewhere in the brain (*Wang, 2010*; *Buzsáki and Wang, 2012*), is ostensibly well-supported by the purported intermittent synchronization of MC firing across a sparse fraction of gamma-frequency cycles (*Bathellier et al., 2006*; *Rojas-Líbano and Kay, 2008*; *Brea et al., 2009*; *Wang, 2010*). However, our results instead show sustained synchronization of periodically firing principal cells – especially TCs – across timeframes consistent with several consecutive gamma-frequency cycles. This evidence of dense synchrony, together with a broader failure of our data to reveal clear temporal gating, motivates reassessment of how well sparsely synchronous neocortical and hippocampal regimes generalize to the MOB.

Examples documenting sparse synchrony in the MOB reveal phase-locked firing of individual MCs within approximately half (*Bathellier et al., 2006*) to two-thirds (*Kashiwadani et al., 1999*) of gamma-frequency cycles. While indeed evincing synchronization of individual MCs on only a subset of oscillatory cycles, this level of synchronization is unequivocally distinct from the Poisson-like phase-locked firing of principal cells within only ~5% of gamma-frequency cycles in neocortex and hippocampus (*Wang, 2010*). MC firing at net rates slower than gamma frequencies both in vitro (*Bathellier et al., 2006*) and in vivo (*Cang and Isaacson, 2003*) has further been taken as indirect evidence of sparse synchrony in the MOB (*Bathellier et al., 2006*; *Brea et al., 2009*). However, instantaneous firing rates within spike clusters – particularly within the timeframe of theta-frequency-nested gamma-frequency oscillations, rather than averaged broadly across seconds following sensory input – do register within gamma frequencies. Indeed, consistent with our in vitro results, TCs and MCs in vivo exhibit highly periodic sensory-evoked firing specifically at fast- and slow-gamma frequencies, respectively (*Margrie and Schaefer, 2003*; *Fukunaga et al., 2014*). Moreover, extracellularly recorded MOB units exhibiting strong sensory-evoked firing at the transition of inhalation-to-exhalation – likely encompassing TCs and strongly activated MCs (*Fukunaga et al., 2012*) – phase-lock to gamma-frequency oscillations in the LFP primarily when firing at gamma frequencies (*Cenier et al., 2009*), with a prevailing 1:1 spike-to-oscillatory cycle relationship (*David et al., 2009*).

The preponderance of data, including our current results, thus most parsimoniously aligns with a densely synchronous regime in which gamma-frequency oscillations emerge from the synchronization of periodically firing resonant neural oscillators. Indeed, periodic optogenetic activation of MOB principal cells (predominantly MCs – see *Arenkiel et al., 2007*; *Lepousez and Lledo, 2013*) at rates spanning 25–90 Hz evokes a peak in MOB gamma-frequency oscillations in vivo specifically when principal cells fire at ~40–60 Hz (*Lepousez and Lledo, 2013*) – results in direct agreement with dense synchronization of resonant neural oscillators and orthogonal to equivalent periodic activation of neocortical principal cells (*Cardin et al., 2009*). Such dense synchrony amid fast network oscillations in the MOB has critical implications for how information is propagated across synapses with frequency-dependent plasticity (*Oswald and Urban, 2012*), another key direction for future research.

While our results provide no indication of a temporal gating mechanism underlying neural synchrony in the MOB, phase-resetting and temporal gating mechanisms are not inherently incompatible (*Li and Cleland, 2017*). Indeed, the reciprocal dendrodendritic architecture of many inhibitory circuits in the MOB will necessarily promote more synchronous and powerful lateral inhibition among synchronously rather than asynchronously firing principal cells (*Marella and Ermentrout, 2010*), suggesting a potential avenue by which synchronous inhibition increasingly gates principal cell firing across time, particularly across slower frequencies (*David et al., 2015*). A critical consideration, however, is that temporal gating mediated specifically by GCs implies phase-locking of GC firing to gamma-frequency oscillations (*Wang, 2010*), which is not observed (*Lagier et al., 2004*) and cannot intuitively be supplanted by highly localized and asynchronous subthreshold release (*Burton, 2017*). The MOB is host to an array of other interneurons capable of mediating differential inhibition among TCs and MCs, however (*Banerjee et al., 2015*; *Burton et al., 2017*; *Geramita and Urban, 2017*; *Liu et al., 2019*), motivating further investigation into how other cell types contribute to fast network oscillations in the MOB.

## Materials and methods
### Animals
All experiments were completed in compliance with the guidelines established by the Institutional Animal Care and Use Committee of the University of Pittsburgh (protocol #18103723). Optogenetic

experiments used gene-targeted OMP-ChR2:EYFP mice (Omp$^{tm1.1(COP4*/EYFP)Tboz}$/J; stock number 014173, The Jackson Laboratory; RRID:IMSR_JAX:014173), in which the endogenous olfactory marker protein (*OMP*) gene is replaced with the H134R variant of channelrhodopsin-2 fused to enhanced yellow fluorescent protein (*ChR2:EYFP*), driving ChR2:EYFP expression in all mature OSNs (*Smear et al., 2011*). OMP-ChR2:EYFP mice were maintained on a C57BL/6J-albino background and used as heterozygotes to minimize OSN signaling deficits, as previously described (*Burton and Urban, 2015*). Mice were socially housed and maintained on a 12 hr light/dark cycle.

## Slice preparation

Postnatal day 20–28 mice (n = 25) of both sexes were anesthetized with isoflurane and decapitated into ice-cold oxygenated dissection solution containing the following (in mM): 125 NaCl, 25 glucose, 2.5 KCl, 25 NaHCO$_3$, 1.25 NaH$_2$PO$_4$, 3 MgSO$_4$, and 1 CaCl$_2$. Brains were isolated and acute horizontal slices (310 µm thick) were prepared using a vibratome (5000mz-2, Campden Instruments Ltd.; or VT1200S, Leica Biosystems). Slices recovered for 30 min in ~37°C oxygenated Ringer's solution that was identical to the dissection solution except with lower Mg$^{2+}$ concentrations (1 mM MgSO$_4$) and higher Ca$^{2+}$ concentrations (2 mM CaCl$_2$). Slices were then stored at room temperature until recording.

## Electrophysiology

Slices were continuously superfused with warmed oxygenated Ringer's solution (temperature measured in bath: 30–32°C). Cells were visualized using infrared differential interference contrast video microscopy. Recordings were targeted to the medial MOB, where the MCL reliably appeared as a uniformly compact cell layer, facilitating the differentiation of cell types. MCs and TCs were differentially targeted based on laminar position of somata within the MCL and EPL, respectively, and confirmed through post-hoc visualization of Neurobiotin labeling, as previously described (*Burton and Urban, 2014*). In particular, MCs were differentiated from deep TCs if >50% of their cell body resided within the MCL. Recovered morphologies are displayed with images collected at a single plane, with long lateral dendrites extending out of focus. Homotypic and heterotypic pairs were differentiated by assessment of spontaneous long-lasting depolarization synchrony in current- and voltage-clamp recordings (*Carlson et al., 2000*) as well as post-hoc Neurobiotin visualization. No difference existed between recording age of MCs (25.2 ± 2.3 days) and TCs (24.6 ± 2.5 days) (Wilcoxon rank-sum test, p=0.24). Cell-attached and current-clamp recordings were made using electrodes (7.7 ± 1.4 MΩ) filled with the following (in mM): 135 K-gluconate, 1.8 KCl, 9 HEPES, 10 Na-phosphocreatine, 4 Mg-ATP, 0.3 Na-GTP, 0.2 EGTA, and 0.025 Alexa Fluor 594 hydrazide (Thermo Fisher Scientific), along with 0.2% Neurobiotin (Vector Labs). In optogenetic experiments, spike timing was recorded in cell-attached mode when possible to both minimize potential disruption of endogenous cellular physiology and to facilitate comparison of results to previous in vivo single-unit recordings (e.g., *Kashiwadani et al., 1999*). In a subset of pairs (n = 8 TC pairs, n = 6 MC pairs), one or both cells spontaneously entered whole-cell mode, and spike timing was consequently recorded in current-clamp mode. Following each cell-attached recording, whole-cell access was obtained to verify stable resting membrane potentials (TC: –57.1 ± 4.0 mV [n = 52]; MC: –51.9 ± 2.3 mV [n = 32]) matching previously observed values (https://neuroelectro.org/neuron/131/ and https://neuroelectro.org/neuron/129/; *Tripathy et al., 2014*; *Tripathy et al., 2015*; RRID:SCR_006274). In a subset of pairs, voltage-clamp recordings of photostimulation-evoked excitatory currents were additionally obtained at a holding potential of –60 mV (i.e., near the reversal potential of IPSCs), while unitary lateral excitation was examined in current-clamp recordings at resting membrane potential. Voltage-clamp recordings of IPSCs were obtained at a holding potential of +10 mV (i.e., near the reversal potential of excitatory postsynaptic currents) using electrodes (8.4 ± 1.3 MΩ) filled with the following (in mM): 123 Cs-gluconate, 3.8 K-gluconate, 1.8 KCl, 9 HEPES, 10 Na-phosphocreatine, 4 Mg-ATP, 0.3 Na-GTP, 4.4 QX-314, 4.4 BAPTA, and 0.025 AF594, along with 0.2% Neurobiotin. This approach to recording IPSCs, while precluding a direct within-pair comparison of synaptic inhibition to spike timing (due to electrode solution differences), provided the greatest resolution of IPSCs while avoiding contaminating excitatory postsynaptic currents. Such consideration is particularly relevant in TC recordings, where excitatory input frequently appears as both long-lasting depolarizing currents as well as more phasic postsynaptic currents with kinetics indistinguishable from IPSCs (data not shown), reflecting the prominent monosynaptic input from OSNs to TCs (*Gire et al., 2012*; *Burton and Urban, 2014*;

*Geramita and Urban, 2017*; *Jones et al., 2020*). BAPTA was included in the electrode solution to minimize the contribution of depolarization-evoked recurrent inhibition to voltage-clamp recordings of IPSCs (*Isaacson and Strowbridge, 1998*). For optogenetic stimulation, slices were illuminated by a 75 W xenon arc lamp passed through a YFP filter set and 60× water-immersion objective centered on the glomerular layer superficial to the recorded pair, with all field-stops fully open. Light power density exiting the objective was 1.2 mW/mm$^2$. The photostimulation protocol consisted of five 10 ms light pulses delivered with an inter-pulse interval of 200 ms (i.e., 5 Hz), except where noted. Electrophysiological data were low-pass filtered at 4 kHz and digitized at 10 kHz using a MultiClamp 700B amplifier (Molecular Devices) and an ITC-18 acquisition board (Instrutech) controlled by custom software written in IGOR Pro (WaveMetrics).

## Data analysis

No differences in patterns of spike timing or synaptic input were observed between sex, and data were therefore pooled across male and female mice. Given the lack of available data on TC synchrony and potential cell-type differences, no a priori power analyses were performed to determine target sample sizes. Experiments were instead designed to encompass a comparable number of pairs as several previous studies of spike-timing computation and coincident inhibition among MOB principal cells (e.g., *Schoppa and Westbrook, 2001*; *Schoppa and Westbrook, 2002*; *Schoppa, 2006*; *Arevian et al., 2008*; *Giridhar et al., 2011*; *Schmidt and Strowbridge, 2014*; *Arnson and Strowbridge, 2017*).

All rates were calculated sequentially between events as the inverse of the inter-event interval, except where noted. Spike times were detected in cell-attached recordings using cell-specific current thresholds of 18–100 pA and in current-clamp recordings using a voltage-derivative threshold of 20 mV/ms, with post-hoc visual confirmation of accurate spike detection in all trials. For analyses of spike- and IPSC-time synchrony, event times were first extracted from recordings and then convolved with a Gaussian kernel (1 ms standard deviation) to account for slight differences in thresholds (for spike-time analysis) and event detection across cells. Trains of convolved event times were then mean-subtracted. Spectral and cross-spectral densities were calculated using Welch's method (100 ms Hamming window, 95% window overlap). For each pair tested, spike-time synchrony and firing periodicity were analyzed across 10.9 ± 3.5 trials; excitatory currents were analyzed across 6.1 ± 1.9 trials; trials with spontaneous bursts of spikes or long-lasting depolarizations immediately preceding photostimulation onset were excluded from analysis. Latency to excitatory input following photostimulation onset was calculated as the first time at which mean baseline-subtracted currents deviated below 2 standard deviations of the baseline current preceding photostimulation.

IPSCs were detected using a standard template-matching function in Axograph (*Clements and Bekkers, 1997*). Given the high IPSC rate and consequent frequent temporal summation observed, a relatively abridged template was applied, with 4.0 ms total duration, 1.0 ms baseline, 0.4 ms rise time constant, 3.0 ms decay time constant, and 0.5 ms minimum event separation. IPSCs were detected with a threshold amplitude of 2.5× the standard deviation of the baseline noise. 20–80% rise times were calculated for each detected IPSC. Decay constants could not be accurately estimated for a large fraction of IPSCs due to the high IPSC rate; therefore, decay constants were estimated by single-exponential fits to the median waveform across all spontaneous or evoked IPSCs recorded in each cell. For each pair, synaptic inhibition was analyzed across 18.0 ± 6.9 trials.

To test for unitary lateral excitation, short step currents evoking a single spike were sequentially injected into each cell of a pair across 40.7 ± 25.7 trials. For each direction, postsynaptic recordings were aligned to the presynaptic spike time using an upsampled 1 MHz sampling rate and absolute spike threshold of –40 mV to facilitate precise spike-time detection. Postsynaptic response amplitudes were then calculated as the maximum depolarization in the mean baseline-subtracted membrane potential within 15 ms of the presynaptic spike time. Electrical coupling coefficients were calculated as the ratio of postsynaptic-to-presynaptic membrane potential change (averaged across 55.4 ± 32.3 trials) following a 200–900 pA hyperpolarizing step-current injection to the presynaptic cell.

Subthreshold intrinsic resonance was analyzed in a previously collected in vitro dataset of TCs (n = 12; age: 16.3 ± 0.8 days) and MCs (n = 10; age: 16.4 ± 1.3 days) (*Burton and Urban, 2014*). STOs were detected using iterative semi-automated ridge detection applied to continuous Morlet wavelet transforms of subthreshold membrane potentials (isolated by linearly interpolating

voltages across spikes), similar to recent in vivo investigation of STOs (*Fourcaud-Trocmé et al., 2018*). Specifically, ridge detection was initialized for each subthreshold response across a broad frequency range (10–150 Hz), using a relaxed ridge threshold to capture all possible continuous STOs (ΔHz/ms ≤30). For each candidate STO, frequency was defined by the corresponding ridge maximum, and duration was defined by the continuous length of the detected ridge (surrounding the maximum) over which the instantaneous frequency deviated <20% from the ridge maximum frequency, ensuring regularity in candidate STO frequency. Candidate STOs < 2 periods in duration were rejected. For each remaining candidate STO, a sinusoid with matching duration was then generated with amplitude equal to the membrane potential standard deviation, offset equal to the membrane potential mean, frequency equal to the ridge maximum, and phase determined by minimizing the sum of the squared error between sinusoid and membrane potential. The candidate STO and least squares estimate-sinusoid fit were then visually inspected, and either rejected (e.g., due to irregular membrane potential fluctuations) or confirmed as an STO. This process was then iterated for the same subthreshold response by progressively refining frequency bounds to identify and visually compare all possible STOs, using both the continuous wavelet transform spectrogram and subthreshold membrane potential as guidance, with a maximum of one STO confirmed per temporal epoch.

For each statistical test, data normality was first determined by the Shapiro–Wilk test, and non-parametric tests applied where appropriate. For visual comparison of normally distributed data, all individual data points are displayed in addition to sample mean and standard errors. For visual comparison of non-normally distributed data, data are displayed as standard boxplots, with data points denoting sample outliers. For statistical tests performed across multiple individual cells or pairs, p-values were corrected for multiple comparisons using the Benjamini-Hochberg procedure to control the false discovery rate and reported as $p_{BH}$. Values in text are reported as mean ± standard deviation. Line plots with shading denote mean ± standard error, except where noted. Single, double, and triple asterisks in figures denote statistical significance at p<0.05, p<0.01, and p<0.001 levels, respectively.

## Acknowledgements

This work was supported by the National Institute on Deafness and Other Communication Disorders grant R01DC016560 (NNU). We thank Greg LaRocca for excellent technical assistance and members of the Urban, Cheetham, and Ermentrout laboratories for helpful discussion.

## Additional information

### Funding

| Funder | Grant reference number | Author |
| --- | --- | --- |
| National Institute on Deafness and Other Communication Disorders | R01DC016560 | Nathaniel N Urban |

The funders had no role in study design, data collection and interpretation, or the decision to submit the work for publication.

### Author contributions

Shawn D Burton, Designed experiments, performed experiments, analyzed data, wrote the paper; Nathaniel N Urban, Designed experiments, analyzed data, wrote the paper

### Author ORCIDs

Shawn D Burton ⬤ http://orcid.org/0000-0002-8907-6487
Nathaniel N Urban ⬤ http://orcid.org/0000-0002-0365-9068

### Ethics

All experiments were completed in compliance with the guidelines established by the Institutional Animal Care and Use Committee of the University of Pittsburgh (protocol #18103723).

Decision letter and Author response
Decision letter https://doi.org/10.7554/eLife.74213.sa1
Author response https://doi.org/10.7554/eLife.74213.sa2

## Additional files

### Supplementary files
- Transparent reporting form
- Source data 1. Spreadsheet of source data for all figures.

### Data availability
All data generated or analyzed during this study are included in the manuscript and supporting files. Source data files have been provided for all figures.

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
