## [Decision Letter]

**Acceptance summary:**

The manuscript describes an in-depth analysis of slice electrophysiology data from mitral and tufted cell pairs which highlights a unique feature of olfactory bulb population activity, that oscillations represent dense as opposed to sparse synchronizations – particularly among tufted cells. The authors include a controlled experimental model to show that intrinsic subthreshold oscillations entrain tufted cell firing at high and low γ frequencies, suggesting that synaptic inhibition is a mechanism through which subthreshold oscillations are synchronized across cells. These findings are intriguing because they show a unique mechanism of oscillations in the olfactory bulb, which suggests a unique role of OB oscillations in encoding olfactory information to the cortex via tufted cells.

**Decision letter after peer review:**

[Editors’ note: the authors submitted for reconsideration following the decision after peer review. What follows is the decision letter after the first round of review.]

Thank you for submitting your work entitled "Cell and circuit origins of fast network oscillations in the mammalian main olfactory bulb" for consideration by *eLife*. Your article has been reviewed by 3 peer reviewers, and the evaluation has been overseen by a Reviewing Editor and a Senior Editor. The reviewers have opted to remain anonymous.

We are sorry to say that, after consultation with the reviewers, we have decided that your work will not be considered further for publication by *eLife*. The reviewers were interested in the topic, but raised a number of issues in their reviews and the subsequent discussion that preclude further consideration of the manuscript at *eLife* in its current form. The concerns included the possibility that the synchronized stimulus may have contributed to the observed results; that the time window chosen for analysis needed to be extended; and there was a general concern that the authors had overstated what could be concluded from the results. In particular two of the reviewers that additional evidence was needed to support some of the major conclusions. The possible effects of subtypes of TCs and the role of gap junctions was also raised by one of the reviewers. The full comments of the reviewers are included below.

*Reviewer 1:*

The manuscript by Burton and Urban represents a thorough and compelling analysis of slice electrophysiology data from mitral and tufted cell pairs that highlight a unique feature of olfactory bulb population activity – that oscillations represent dense as opposed to sparse synchronizations – particularly among tufted cells. The authors also include data with new analysis from previous single cell current clamp data to show that intrinsic subthreshold oscillations entrain tufted cell firing at high and low γ frequencies, and suggest that synaptic inhibition is a mechanism through which subthreshold oscillations are synchronized across cells via phase resetting. The findings are intriguing because they provide a new perspective of what influences/drives oscillations in the olfactory bulb, which in turn, suggests a unique role of OB oscillations in encoding olfactory information and transmission of that information to the cortex via tufted cells.

While the analyses are thorough, well-presented in the figures, and explained in detail throughout the text, some of the claims of state dependence and phase resetting via synaptic inhibition and/or intrinsic properties may be over-stated without directly testing through experimental manipulations, or via modeling impacts of various forms of synaptic inhibition and/or intrinsic channel/current properties. Overall, however, although the mechanistic and causality claims could be tempered, this is a valuable and high-quality dissection of temporal firing properties between projection cell types in the bulb, with a scholarly and insightful perspective.

– In the discussion the authors conclude that their results provide evidence that link TCs and MCs to state-dependent γ oscillations. However, these data do not directly address state-dependence, and thus seems to be somewhat intriguing speculation. This could be tested with additional experiments, for example including neuromodulator agonists and antagonists, but if not tested should at least be buffered in concept without such data.

– Similarly, the authors imply that their data show that synchronous inhibition is responsible for synchronizing TCs through phase resetting. Care should be taken that this not be considered a concrete conclusion given that the current data do not show that synchronous inhibition resets TC STO phase.

Both of the above points could be addressed simply by tempering language and being more explicit about what is directly shown, and what is speculation. Alternatively, the second point might actually be tested with modeling. In fact, modeling may be an avenue to substantiate both the IPSC and STO analyses, as well as the capacity for different potassium/sodium currents to influence intrinsic properties.

– The magnitude of the CPSD suffers a similar artifact to the cross correlogram and would benefit from a similar subtraction of simulated correlations. Since many analyses are focused on the CPSD and CPSD ridges, it is important to see that these hold up even after subtraction of artifactual CPSD.

– More of a curiosity rather than concern: would there be any reason to question if homotypic TC/MC pairs would be more synchronized over MC pairs?

*Reviewer #2:*

This manuscript describes studies in brain slices that examine the mechanisms of γ frequency synchronized oscillations in the main olfactory bulb (MOB), specifically comparing these oscillations between the two output cell-types, the mitral cells (MCs) and tufted cells (TCs). The authors' main conclusions include that γ synchrony is both larger in magnitude and also faster in frequency in TCs than in MCs. The greater synchrony in TCs is attributed to their greater intrinsic resonance rather than differences in synaptic connections. The authors also conclude that TCs, due to their greater resonance, are the primary driver of synchronized γ oscillations in MOB.

The question of what drives synchronized γ oscillations in MCs of MOB has received considerable attention in the past two decades, but γ oscillations in TCs have not been mechanistically analyzed. Also, there is still disagreement in the field of what drives γ oscillations in MCs, with different studies placing varying levels of emphasis on the role of synchronized inhibitory synaptic inputs versus intrinsic cell resonance. Thus, the study is addressing important questions, and the results, if conclusive, could have a major impact in the olfactory field as well as the understanding of γ oscillations in the brain more generally. The experiments appear to be of high quality and the results are also generally well-presented. Some of the experimental results are also convincing, for example the pair-cell recordings that show that TCs and MCs do not have major differences in their inhibitory input that could explain differences in the γ oscillations.

I do however have a number significant concerns with the analysis and interpretation of results. One key overall issue is that the work, in my view, does not adequately exclude the possibility that many of the observations are due to the strong optogenetic stimulus. This concern applies both to their studies showing that TCs have more synchronized spiking than MCs (Figure 1) and also their evidence that the greater intrinsic resonance in TCs is responsible for the greater oscillatory synchrony in TCs (Figure 7). TCs display a strong, rapid phasic response to the stimulus (Figure 1), which is concerning in this respect. There are also concerns with the conclusion that TCs are the primary driver of synchronized spiking overall in MOB. While the authors may be able to show that TCs have more oscillatory synchrony than MCs (assuming caveats are addressed), such a result does not imply that TCs are driving the synchronized oscillations in MCs.

While the study is addressing important questions and some of the results are convincing, I have the following specific concerns.

1. The authors have not adequately addressed the possibility that the high level of fast synchrony in TCs does not simply reflect the response to the strong synchronized, optogenetic stimulus. The issue of course is that each cell might expected to have an initial synchronized depolarization immediately after the stimulation of OSNs. This could make the first spikes synchronized and perhaps also a few additional, later spikes if the cells have AHPs that are similar in duration. Based on the spike frequency plots in Figure 1G and 1J, TCs appear to display a strong fast phasic response to the stimulus, raising the possibility that at least some of the early spikes could be synchronized by the stimulus. There are other results as well that contribute to this concern, including the fact that MCs, which display no or little fast phasic spike response (Figure 1A, 1D), have much less spike synchrony on a fast time-scale.

The authors attempt to address this concern in part by subtracting out cross-correlograms generated from simulated data from their experimental cross-correlogams. They indicate that the cross-correlograms from the simulated data did not display fast time-scale synchrony, which they take as evidence (I believe) that the stimulus is not driving the fast synchrony. This is somewhat ambiguous to me however, and it's not clear that their method would have accounted for artifacts that could have arisen due to the somewhat imprecise nature of optogenetic stimulation. The resulting trial-to-trial variation in OSN stimulation time could have caused the first spike(s) in the two cells for a given trial to be more synchronized than spikes taken from a data set in which trial number has been randomized. In this scenario, the cross-correlogams generated from the simulated data may lack a fast peak, yet the cause of the rapid synchrony would still be the stimulus.

It would be more convincing if the authors simply showed that the pair-cell recordings displayed just as much spike synchrony in the latter part of each stimulus epoch (e.g., for spikes recorded 100-200 ms after the stimulus) as compared to just after each stimulus. The authors would additionally need to show that TCs showed more synchrony than MCs during the latter 100-200 ms period after the stimulus.

(2) The case that the greater oscillatory synchrony in TCs is due to their greater intrinsic resonance could be more strongly made. Here one issue of course is whether the subthreshold oscillations due to intrinsic cell properties that can be seen quite clearly when the TCs are directly depolarized (in Figure 6) also have an impact under the much noisier conditions following OSN stimulation. Although it may be difficult, it would be more convincing if the authors can show more directly that the subthreshold oscillations exist following OSN stimulation. An analysis similar to that shown for the direct depolarization condition in Figure 6 could be conducted.

Similar to what was discussed in Major Point 1 related to synchronized spiking, there is also a concern that the strong synchronized oscillations in TCs that they attribute to resonance could be explained by the stimulus. Do the authors obtain the same results if they restrict the analysis performed in Figure 7 to the last 100 ms of the 200-ms epoch following the start of each stimulus?

(3) The authors' conclusion that TCs are the major driver of oscillatory synchrony in MOB does not appear to be well-supported. This conclusion appears to be based just on the fact that TC oscillations are larger than those of MCs, but this does not at all mean that TCs are driving the MC oscillations (which is what I believe the authors are implying). The oscillations in the two cell types could of course be driven by completely independent mechanisms. In my view, the present study does not provide substantial evidence (direct or indirect) that TCs are driving the oscillatory synchrony in MCs.

(4) A final concern, somewhat less central but still notable, has to do with the author's conclusion that the TC oscillatory synchrony occurs at a higher frequency than that of MCs. One issue is that MC synchrony in their experiments is very small in scale, which makes it difficult to analyze. When the cell-pairs are analyzed individually (Figure 1-Supp. Figure 2B), the results of only 4 MC pairs are plotted. Moreover, when comparing the ridge frequencies of those 4 MC pairs with those of the TC pairs, there is no clear evidence for differences. Only 1 out of 4 MCs display very high ridge frequencies (>80 Hz), but the fraction of TCs that display such high ridge frequencies is similarly low. A significant difference is observed in the ridge frequencies between MCs and TCs when they examine their results across their cell populations (Figure 1Q), yet it is somewhat difficult to know what this means when differences cannot be observed when MC pairs are analyzed individually.

That oscillatory synchrony in MCs is small in magnitude – and thus difficult to analyze – is also reinforced from the simple spike correlograms that the authors show for MCs (Figure 1 —figure supplement 1). The authors indicate in the main text that 50% of MC pairs displayed evidence for synchronized spikes, as reflected by an R(spike) value > 1.2, yet the spike correlograms are noisy enough (due to a low number of events) that it is not clear that such R(spike) values are significant.

*Reviewer #3:*

Burton and Urban investigated the olfactory nerve (ON) stimulation-evoked spike-time synchrony among the two major types of principal output neurons of the main olfactory bulb (MOB) – tufted cells (TCs) and mitral cells (MCs) in slice preparations with the objective to understand the cellular and circuit mechanisms underlying the sensory-evoked γ oscillations, which reflect synchronized population activities of output neurons and have been widely observed in both fast and slow γ frequency bands in the MOB with in vivo extracellular local field potential (LFP) recordings. The methodological strength of this study include: (1) ON was optogenetically activated with minimal stimulus artifact detected in the recorded neurons such that the peak time of all spikes in the recorded neurons can be accurately measured and included for analysis; (2) firing spikes in pairs of TCs or MCs were recorded with the cell-attached mode, an extracellular recording approach at the single cell level enabling direct comparison of data with those collected with in vivo single-unit recordings in the literature. Based on these rigorous experimental approaches, they found that multiglomerular activation results in more rapid, widespread, and precise periodic synchronization of ring activities in TCs compared to MCs. Moreover, TC synchrony extended across fast-γ frequencies and showed a sweeping deceleration, which directly mirror in vivo extracellular recordings in the MOB, while MC synchrony was limited to slow-γ frequencies and less dynamic. These findings support a central role of TCs and related circuits in the generation of sensory-evoked MOB γ oscillations (especially at the fast frequency band), which are usually attributed to the synchronized MC activities temporally gated by the granule cell (GC)-mediated lateral inhibition. Further analyses suggest that differences in periodic spike time synchrony between the two major types of MOB output neurons attribute to their differences in intrinsic resonance that is more widespread and better entrains firing among TCs than MCs. No significant differences were revealed in the synchronization of inhibitory synaptic inputs among TCs vs MCs, indicating that γ oscillations are not generated due to lateral inhibition temporally gated by activities of GCs as previously perceived. Instead, the author found that inhibitory synaptic input played a key role in resetting the intrinsic resonance and phase-locked firing activities in both TCs and MCs. Overall, this work presents solid data with thorough analyses supporting their hypothesis that TCs and related circuits are the major origin of sensory-triggered γ oscillations (especially the fast frequency ones) in the MOB, challenging the classic view of MCs with the GC-mediated lateral inhibition as the main contributors.

Please address the following concerns

1. TCs in the MOB are classifies into multiple subtypes including superficial (sTCs), middle (mTCs), and deep (dTCs) tufted cells as targeted by the authors in this manuscript. Due to their different somatic locations in the EPL, the lateral dendrites of these TC subtypes are distributed to distinct laminar portions (superficial to deep) of the EPL where they very likely receive inhibitory synaptic input from different populations (superficial or deep) of GCs. Thus, the weaker synchrony of inhibitory synaptic input in the γ band among different subtypes of TCs compared to MCs as shown in Figure 5 might be because these TCs receive inhibitory input from distinct GCs while MCs may more likely receive inhibitor input from the same subpopulation of GCs.

2. Gap junctions are widely present in the MOB glomerular circuit including among MCs and TCs. The authors did not mention any potential roles of this electrical synaptic communication in spike time synchrony among MCs or TCs. Actually, this type of work to answer this question could have been done in experiments shown in Figure 5.

3. Evidence shows chemical synaptic connections among apical dendrites of MCs and TCs or between TCs and MCs. What roles do these dendrodendritic synaptic transmissions potentially play in spike time synchrony among TCs or MCs?

4. Please provide more details on how dTCs and MCs are preselected and differentiated. Authors described in the Method section that cell-types were identified by their somatic location in the laminar layer of the MOB. But the cell body locations of dTCs and MCs are practically difficult to determine and differentiate in slice preparations.

5. MCs in Figure 1A do not look like heterotypic. Please replace it with a better reconstruction photo.

---

## [Author Response]

[Editors’ note: the authors resubmitted a revised version of the paper for consideration. What follows is the authors’ response to the first round of review.]

Reviewer 1:The manuscript by Burton and Urban represents a thorough and compelling analysis of slice electrophysiology data from mitral and tufted cell pairs that highlight a unique feature of olfactory bulb population activity – that oscillations represent dense as opposed to sparse synchronizations – particularly among tufted cells. The authors also include data with new analysis from previous single cell current clamp data to show that intrinsic subthreshold oscillations entrain tufted cell firing at high and low γ frequencies, and suggest that synaptic inhibition is a mechanism through which subthreshold oscillations are synchronized across cells via phase resetting. The findings are intriguing because they provide a new perspective of what influences/drives oscillations in the olfactory bulb, which in turn, suggests a unique role of OB oscillations in encoding olfactory information and transmission of that information to the cortex via tufted cells.While the analyses are thorough, well-presented in the figures, and explained in detail throughout the text, some of the claims of state dependence and phase resetting via synaptic inhibition and/or intrinsic properties may be over-stated without directly testing through experimental manipulations, or via modeling impacts of various forms of synaptic inhibition and/or intrinsic channel/current properties. Overall, however, although the mechanistic and causality claims could be tempered, this is a valuable and high-quality dissection of temporal firing properties between projection cell types in the bulb, with a scholarly and insightful perspective.-In the discussion the authors conclude that their results provide evidence that link TCs and MCs to state-dependent γ oscillations. However, these data do not directly address state-dependence, and thus seems to be somewhat intriguing speculation. This could be tested with additional experiments, for example including neuromodulator agonists and antagonists, but if not tested should at least be buffered in concept without such data.

We agree that this conclusion in the original manuscript was speculative and not explicitly supported by the data. While our experimental platform provides an exciting opportunity for exploring the state dependence of fast network oscillations through pharmacological manipulation, as the reviewer suggests, such investigation would be best served as its own study. We have thus removed this conclusion from our new submission.

– Similarly, the authors imply that their data show that synchronous inhibition is responsible for synchronizing TCs through phase resetting. Care should be taken that this not be considered a concrete conclusion given that the current data do not show that synchronous inhibition resets TC STO phase.Both of the above points could be addressed simply by tempering language and being more explicit about what is directly shown, and what is speculation. Alternatively, the second point might actually be tested with modeling. In fact, modeling may be an avenue to substantiate both the IPSC and STO analyses, as well as the capacity for different potassium/sodium currents to influence intrinsic properties.

We agree that this conclusion in the original manuscript was speculative and not explicitly supported by the data. In particular, while the sum of our evidence strongly points toward a phase-resetting mechanism of γ-frequency spike-time synchrony, similar to mechanisms we have previously proposed (Galán et al., 2006), we did not directly demonstrate that neurons are synchronized by synaptic inhibition resetting their periodic firing phase. This lack of direct demonstration in part reflects the majority of our data comprising cell-attached recordings, and in part reflects the difficulty of pinpointing such causality, which may be best accomplished in a complementary study combining modeling and pharmacological manipulation, as the reviewer suggests. We have thus amended the conclusions throughout our new submission to more clearly state what the data directly demonstrates, and what may be likely as a result. For example:

“Collectively, our results thus argue that synchronization of periodically-firing TCs, likely mediated by a phase-resetting mechanism, strongly contributes to fast network oscillations in the MOB.” (Discussion, lines 509-510)

In addition, in focusing primarily on the dense γ-frequency synchronization of tufted cells, we have substantially restructured our new submission such that examination of subthreshold oscillations is only presented as a complementary assessment of oscillatory behavior/tendency:

“That TCs exhibit greater oscillatory behavior than MCs despite comparable synaptic input following multiglomerular activation suggests that the intrinsic biophysical properties of TCs yield greater tendency toward oscillatory behavior (i.e., resonance) than the biophysical properties of MCs. Therefore, to begin to trace the cell-type differences in periodic spike-time synchrony to potential biophysical sources, in our final set of analyses we examined subthreshold oscillations (STOs) – a manifestation of intrinsic resonance (Hutcheon and Yarom, 2000) – among MCs and TCs.” (Results, lines 447-452)

– The magnitude of the CPSD suffers a similar artifact to the cross correlogram and would benefit from a similar subtraction of simulated correlations. Since many analyses are focused on the CPSD and CPSD ridges, it is important to see that these hold up even after subtraction of artifactual CPSD.

As suggested, in our new submission we show CPSD analysis of experimental and simulated spike times, as well as the subtraction of simulated spike-time CPSD spectrograms from experimental spike-time CPSD spectrograms (Figure 1 —figure supplement 2). Importantly, this analysis shows that the γ-frequency spike-time synchrony observed among mitral cells and especially among tufted cells well exceeds chance levels of synchrony observed among simulated spike times. Equivalent subtraction procedures were likewise applied to the analysis of IPSC-time synchrony (Figure 6 —figure supplement 4) and analysis of firing periodicity (Figure 7 —figure supplement 1).

– More of a curiosity rather than concern: would there be any reason to question if homotypic TC/MC pairs would be more synchronized over MC pairs?

Unfortunately, our current data provide limited insight into interactions between mitral and tufted cells, particularly in comparison to interactions between mitral cells. However, we agree that this topic is of great interest and merits further thought and investigation. Toward this end, we have included the following discussion in our new submission:

“How synchronization of TC firing influences the activity of MCs linked to the same glomerulus, or even whether multiglomerular activation can synchronize MC firing to TC firing, likewise remain open questions of outstanding interest. TCs can laterally excite MCs linked to the same glomerulus (Pimentel and Margrie, 2008; Najac et al., 2011), which not only underscores the surprising absence of lateral excitation between homotypic TCs, but also suggests that homotypic TC-MC pairs may parallel homotypic MC pairs in exhibiting synchronous irregular firing but more limited γ-frequency synchrony. Consistent with this prediction, spontaneous firing within homotypic TC-MC pairs exhibits less precise spike-time synchrony than spontaneous firing within homotypic TC pairs (Ma and Lowe, 2010).” (Discussion, lines 602-610)

Reviewer #2:[…] While the study is addressing important questions and some of the results are convincing, I have the following specific concerns.1. The authors have not adequately addressed the possibility that the high level of fast synchrony in TCs does not simply reflect the response to the strong synchronized, optogenetic stimulus. The issue of course is that each cell might expected to have an initial synchronized depolarization immediately after the stimulation of OSNs. This could make the first spikes synchronized and perhaps also a few additional, later spikes if the cells have AHPs that are similar in duration. Based on the spike frequency plots in Figure 1G and 1J, TCs appear to display a strong fast phasic response to the stimulus, raising the possibility that at least some of the early spikes could be synchronized by the stimulus. There are other results as well that contribute to this concern, including the fact that MCs, which display no or little fast phasic spike response (Figure 1A, 1D), have much less spike synchrony on a fast time-scale.The authors attempt to address this concern in part by subtracting out cross-correlograms generated from simulated data from their experimental cross-correlogams. They indicate that the cross-correlograms from the simulated data did not display fast time-scale synchrony, which they take as evidence (I believe) that the stimulus is not driving the fast synchrony. This is somewhat ambiguous to me however, and it's not clear that their method would have accounted for artifacts that could have arisen due to the somewhat imprecise nature of optogenetic stimulation. The resulting trial-to-trial variation in OSN stimulation time could have caused the first spike(s) in the two cells for a given trial to be more synchronized than spikes taken from a data set in which trial number has been randomized. In this scenario, the cross-correlogams generated from the simulated data may lack a fast peak, yet the cause of the rapid synchrony would still be the stimulus.It would be more convincing if the authors simply showed that the pair-cell recordings displayed just as much spike synchrony in the latter part of each stimulus epoch (e.g., for spikes recorded 100-200 ms after the stimulus) as compared to just after each stimulus. The authors would additionally need to show that TCs showed more synchrony than MCs during the latter 100-200 ms period after the stimulus.

We thank the reviewer for raising this important issue. In our new submission, we explicitly present this potential caveat and provide several lines of evidence, including the suggested analysis of spike-time synchrony within consecutive time windows, as to why we believe our data reflects real cell-type differences in network-driven synchronization rather than an artificially strong activation of tufted cell sensory input:

“As a caveat, it is possible that the differences observed in TC vs. MC spike-time synchrony reflect the artificial conditions of our experimental preparation, rather than cell-type differences in network-driven synchronization poised to shape sensory processing in vivo. […] Attenuation of our optogenetic stimulus by limited light penetrance into the tissue contributed to such gradual glomerular activation, with photostimulation routinely failing to activate glomeruli deep in the slice (data not shown). Excitatory input was also completely devoid of any γ-frequency patterning (Figure 2 —figure supplement 1F), further arguing that the periodic spike-time synchrony observed was not directly driven by the stimulus.” (Results, lines 157-191)

(2) The case that the greater oscillatory synchrony in TCs is due to their greater intrinsic resonance could be more strongly made. Here one issue of course is whether the subthreshold oscillations due to intrinsic cell properties that can be seen quite clearly when the TCs are directly depolarized (in Figure 6) also have an impact under the much noisier conditions following OSN stimulation. Although it may be difficult, it would be more convincing if the authors can show more directly that the subthreshold oscillations exist following OSN stimulation. An analysis similar to that shown for the direct depolarization condition in Figure 6 could be conducted.Similar to what was discussed in Major Point 1 related to synchronized spiking, there is also a concern that the strong synchronized oscillations in TCs that they attribute to resonance could be explained by the stimulus. Do the authors obtain the same results if they restrict the analysis performed in Figure 7 to the last 100 ms of the 200-ms epoch following the start of each stimulus?

As noted by both Reviewer 1 and Reviewer 2, our original conclusion that coincident synaptic inhibition acts upon subthreshold oscillations within tufted cells to synchronize periodic gamma frequency firing through a phase-resetting mechanism was an interpretation and not explicitly demonstrated by the data. As stated above, this lack of explicit demonstration in part reflects the majority of our data comprising cell-attached recordings, and in part reflects the difficulty of pinpointing such causality without the ability to selectively manipulate synaptic synchrony and/or subthreshold oscillations. Review of the minority of tufted cell pairs in which spiking activity evoked by multiglomerular activation was recorded in current-clamp mode further revealed either essentially continuous periodic firing or periodic firing interposed with subthreshold epochs strongly contaminated by synaptic activity, precluding the direct analysis of subthreshold oscillations suggested by the reviewer. We have thus amended the conclusions throughout our new submission to more clearly state what the data directly demonstrates, and what may be likely as a result. For example:

“Collectively, our results thus argue that synchronization of periodically-firing TCs, likely mediated by a phase-resetting mechanism, strongly contributes to fast network oscillations in the MOB.” (Discussion, lines 509-510)

Moreover, we have substantially restructured our new submission such that examination of subthreshold oscillations is only presented as a complementary assessment of oscillatory behavior/tendency:

“That TCs exhibit greater oscillatory behavior than MCs despite comparable synaptic input following multiglomerular activation suggests that the intrinsic biophysical properties of TCs yield greater tendency toward oscillatory behavior (i.e., resonance) than the biophysical properties of MCs. Therefore, to begin to trace the cell-type differences in periodic spike-time synchrony to potential biophysical sources, in our final set of analyses we examined subthreshold oscillations (STOs) – a manifestation of intrinsic resonance (Hutcheon and Yarom, 2000) – among MCs and TCs.” (Results, lines 447-452)

Finally, we have analyzed firing periodicity throughout the full 200-ms photostimulation cycle in our new submission (Figure 7 —figure supplement 3), as suggested by the reviewer. Importantly, these results show that the greater firing periodicity (i.e., oscillatory behavior) of tufted cells than mitral cells persists throughout the full 200 ms, suggesting that the prominent oscillatory behavior of tufted cells is not directly driven by an artificially strong optogenetic stimulus. Moreover, as stated in the Discussion of our new submission, these results further agree with the highly periodic fast-γ-frequency firing observed among TCs in vivo (Fukunaga et al., 2014).

(3) The authors' conclusion that TCs are the major driver of oscillatory synchrony in MOB does not appear to be well-supported. This conclusion appears to be based just on the fact that TC oscillations are larger than those of MCs, but this does not at all mean that TCs are driving the MC oscillations (which is what I believe the authors are implying). The oscillations in the two cell types could of course be driven by completely independent mechanisms. In my view, the present study does not provide substantial evidence (direct or indirect) that TCs are driving the oscillatory synchrony in MCs.

We strongly agree with the reviewer, and have amended the imprecise wording in our original submission. Our intent was to convey only that synchronization of tufted cell firing strongly contributes to fast network oscillations in the olfactory bulb. We have thus amended the conclusions throughout our new submission to more clearly convey this finding. Moreover, we now explicitly state:

“How synchronization of TC firing influences the activity of MCs linked to the same glomerulus, or even whether multiglomerular activation can synchronize MC firing to TC firing, likewise remain open questions of outstanding interest.” (Discussion, lines 602-604)

(4) A final concern, somewhat less central but still notable, has to do with the author's conclusion that the TC oscillatory synchrony occurs at a higher frequency than that of MCs. One issue is that MC synchrony in their experiments is very small in scale, which makes it difficult to analyze. When the cell-pairs are analyzed individually (Figure 1-Supp. Figure 2B), the results of only 4 MC pairs are plotted. Moreover, when comparing the ridge frequencies of those 4 MC pairs with those of the TC pairs, there is no clear evidence for differences. Only 1 out of 4 MCs display very high ridge frequencies (>80 Hz), but the fraction of TCs that display such high ridge frequencies is similarly low. A significant difference is observed in the ridge frequencies between MCs and TCs when they examine their results across their cell populations (Figure 1Q), yet it is somewhat difficult to know what this means when differences cannot be observed when MC pairs are analyzed individually.That oscillatory synchrony in MCs is small in magnitude – and thus difficult to analyze – is also reinforced from the simple spike correlograms that the authors show for MCs (Figure 1 —figure supplement 1). The authors indicate in the main text that 50% of MC pairs displayed evidence for synchronized spikes, as reflected by an R(spike) value > 1.2, yet the spike correlograms are noisy enough (due to a low number of events) that it is not clear that such R(spike) values are significant.

Given the limited occurrence and precision of mitral cell synchrony in our dataset, our new submission focuses more on the widespread and robust tufted cell spike-time synchrony across fast- and slowgamma frequency oscillations than on the frequency differences between MC and TC synchrony. We believe this new focus is not only more concretely supported by the data, as the reviewer notes, but also still highlights a novel and important finding. Moreover, we now explicitly describe the issue raised by the reviewer as an important caveat to consider:

“As a caveat, the limited spike-time synchrony detected among MCs in our dataset constrains extensive characterization of the frequency of periodic MC synchronization. Our results thus do not exclude a contribution of MC spike-time synchrony to fast-γ-frequency oscillations in the MOB. However, our results do definitively identify robust periodic spike-time synchrony among TCs as a major contributor to fast- and slow-γ frequency oscillations in the MOB.” (Results, lines 255-260)

In specific response to the concerns regarding the per-pair plot of spike-time CPSD ridge attributes (original submission: Figure 1 —figure supplement 2B; new submission: Figure 1 —figure supplement 3A): this plot was not clearly described in our original submission. In this plot, each point denotes the ridge attributes detected within each of the MC pairs and TC pairs exhibiting spike-time CPSD ridges above the established threshold. Thus, in our original submission, this included only 4 of the 14 MC pairs, as spike-time CPSD ridges were detected in only a small fraction of all MC pairs recorded. In our new submission (containing additional data), this plot now includes 5 of 16 MC pairs and 24 of 26 TC pairs and is more clearly captioned:

“Distribution of ridge CPSD vs. frequency for the 5/16 MC pairs and 24/26 TC pairs exhibiting spiketime CPSD ridges. Ellipses denote mean ± standard deviation of CPSD and frequencies across all ridges detected in each pair.” (Figure 1 —figure supplement 3 caption, lines 1169-1170)

In addition to clearer description, we have also reformatted the plot itself for greater visual clarity. We believe the plot now more clearly shows strong cell-type differences in periodic synchrony: 4 of the 5 MC pairs plotted are concentrated in low-γ frequencies, in strong contrast to the numerous TC pairs concentrated throughout fast-γ frequencies. Both MCs and TCs also have a single pair exhibiting periodic synchrony at frequencies exceeding fast-γ frequencies, as noted by the reviewer and underscoring the now explicitly stated caveat.

Finally, given the limited resolution of the *R_spike_* metric in quantifying periodic synchrony across faster frequencies, we have removed this analysis from our new submission and instead focused on cross correlogram and CPSD analyses.

Reviewer #3:Burton and Urban investigated the olfactory nerve (ON) stimulation-evoked spike-time synchrony among the two major types of principal output neurons of the main olfactory bulb (MOB) – tufted cells (TCs) and mitral cells (MCs) in slice preparations with the objective to understand the cellular and circuit mechanisms underlying the sensory-evoked γ oscillations, which reflect synchronized population activities of output neurons and have been widely observed in both fast and slow γ frequency bands in the MOB with in vivo extracellular local field potential (LFP) recordings. The methodological strength of this study include: (1) ON was optogenetically activated with minimal stimulus artifact detected in the recorded neurons such that the peak time of all spikes in the recorded neurons can be accurately measured and included for analysis; (2) firing spikes in pairs of TCs or MCs were recorded with the cell-attached mode, an extracellular recording approach at the single cell level enabling direct comparison of data with those collected with in vivo single-unit recordings in the literature. Based on these rigorous experimental approaches, they found that multiglomerular activation results in more rapid, widespread, and precise periodic synchronization of ring activities in TCs compared to MCs. Moreover, TC synchrony extended across fast-γ frequencies and showed a sweeping deceleration, which directly mirror in vivo extracellular recordings in the MOB, while MC synchrony was limited to slow-γ frequencies and less dynamic. These findings support a central role of TCs and related circuits in the generation of sensory-evoked MOB γ oscillations (especially at the fast frequency band), which are usually attributed to the synchronized MC activities temporally gated by the granule cell (GC)-mediated lateral inhibition. Further analyses suggest that differences in periodic spike time synchrony between the two major types of MOB output neurons attribute to their differences in intrinsic resonance that is more widespread and better entrains firing among TCs than MCs. No significant differences were revealed in the synchronization of inhibitory synaptic inputs among TCs vs MCs, indicating that γ oscillations are not generated due to lateral inhibition temporally gated by activities of GCs as previously perceived. Instead, the author found that inhibitory synaptic input played a key role in resetting the intrinsic resonance and phase-locked firing activities in both TCs and MCs. Overall, this work presents solid data with thorough analyses supporting their hypothesis that TCs and related circuits are the major origin of sensory-triggered γ oscillations (especially the fast frequency ones) in the MOB, challenging the classic view of MCs with the GC-mediated lateral inhibition as the main contributors.Please address the following concerns1. TCs in the MOB are classifies into multiple subtypes including superficial (sTCs), middle (mTCs), and deep (dTCs) tufted cells as targeted by the authors in this manuscript. Due to their different somatic locations in the EPL, the lateral dendrites of these TC subtypes are distributed to distinct laminar portions (superficial to deep) of the EPL where they very likely receive inhibitory synaptic input from different populations (superficial or deep) of GCs. Thus, the weaker synchrony of inhibitory synaptic input in the γ band among different subtypes of TCs compared to MCs as shown in Figure 5 might be because these TCs receive inhibitory input from distinct GCs while MCs may more likely receive inhibitor input from the same subpopulation of GCs.

In our new submission, we test whether within-pair differences in TC somatic depth correlate with either cross-correlogram or CPSD analyses of IPSC-time synchrony, and observe no clear relationship (Figure 6 —figure supplement 7). Our results thus suggest that differences in TC subtype among paired recordings do not contribute to the overall minimal levels of γ-frequency IPSC-time synchrony recorded. We likewise include equivalent analysis across measures of spike-time synchrony, and again observe no clear relationship (Figure 3 —figure supplement 2). In both of these figures, we additionally plot the depth of each recorded TC to more transparently document the specific TC subtypes recorded.

2. Gap junctions are widely present in the MOB glomerular circuit including among MCs and TCs. The authors did not mention any potential roles of this electrical synaptic communication in spike time synchrony among MCs or TCs. Actually, this type of work to answer this question could have been done in experiments shown in Figure 5.3. Evidence shows chemical synaptic connections among apical dendrites of MCs and TCs or between TCs and MCs. What roles do these dendrodendritic synaptic transmissions potentially play in spike time synchrony among TCs or MCs?

To investigate the role of intraglomerular signaling, particularly among principal tufted cells (in which lateral excitation was previously unexplored), we have performed additional recordings to bring our total number of homotypic tufted cell pairs and homotypic mitral cell pairs to 11 and 4, respectively.

With the addition of this new data, we find lateral excitation (typically asymmetric) among all homotypic mitral cell pairs assayed, matching previous reports. Surprisingly, however, lateral excitation was completely absent among homotypic tufted cells, even while mitral cell pairs and tufted cell pairs exhibited comparable electrical coupling. Thus, the stronger γ-frequency spike-time synchrony observed among tufted cells emerges independent of intraglomerular lateral excitation. Moreover, given the stronger γ-frequency spike-time synchrony observed among homotypic than heterotypic pairs in our dataset, independent of cell type, our results now further suggest that some other mode of intraglomerular signaling – likely including electrical coupling – enhances fast network oscillations in the olfactory bulb. These results are now extensively documented in our new submission (RESULTS, lines 198-233; Figure 3). In addition, we discuss the circuit and computational implications of these results in a new section of the DISCUSSION: “Mechanisms and functions of intraglomerular enhancement of spike-time synchrony” (Discussion, lines 563-610).

4. Please provide more details on how dTCs and MCs are preselected and differentiated. Authors described in the Method section that cell-types were identified by their somatic location in the laminar layer of the MOB. But the cell body locations of dTCs and MCs are practically difficult to determine and differentiate in slice preparations.

In our new submission, we more clearly describe:

“Recordings were targeted to the medial MOB, where the MCL reliably appeared as a uniformly compact cell layer, facilitating the differentiation of cell types. MCs and TCs were differentially targeted based on laminar position of somata within the MCL and EPL, respectively, and confirmed through post-hoc visualization of Neurobiotin labeling, as previously described (Burton and Urban, 2014). In particular, MCs were differentiated from deep TCs if >50% of their cell body resided within the MCL.” (Materials And Methods, lines 729-734).

In addition, Figure 3 —figure supplement 2 and Figure 6 —figure supplement 7 now explicitly plot the depth of each recorded TC.

5. MCs in Figure 1A do not look like heterotypic. Please replace it with a better reconstruction photo.

We have replaced the image in Figure 1A with a new image more clearly resolving the elaboration of the apical dendritic tufts in nearby but distinct glomeruli.